# Structural insights into GrpEL1-mediated nucleotide and substrate release of human mitochondrial Hsp70

Marc A. Morizono ⓘ, Kelly L. McGuire, Natalie I. Birouty & Mark A. Herzik Jr. ⓘ ✉

Maintenance of protein homeostasis is necessary for cell viability and depends on a complex network of chaperones and co-chaperones, including the heat-shock protein 70 (Hsp70) system. In human mitochondria, mitochondrial Hsp70 (mortalin) and the nucleotide exchange factor (GrpEL1) work synergistically to stabilize proteins, assemble protein complexes, and facilitate protein import. However, our understanding of the molecular mechanisms guiding these processes is hampered by limited structural information. To elucidate these mechanistic details, we used cryoEM to determine structures of full-length human mortalin-GrpEL1 complexes in previously unobserved states. Our structures and molecular dynamics simulations allow us to delineate specific roles for mortalin-GrpEL1 interfaces and to identify steps in GrpEL1-mediated nucleotide and substrate release by mortalin. Subsequent analyses reveal conserved mechanisms across bacteria and mammals and facilitate a complete understanding of sequential nucleotide and substrate release for the Hsp70 chaperone system.

Cells maintain the functional states, relative abundance, and proper localization of their myriad proteins through a complex regulatory network governing all aspects of protein synthesis, folding, quality control, and degradation that is collectively termed proteostasis[1,2]. Molecular chaperone systems like heat-shock protein 70 (Hsp70) and Hsp90 proteins play key roles in proteostasis by mediating the initial folding or refolding of many proteins, regulating their biochemical activities, and guiding them to their appropriate destinations in the cell[3,4]. Typically, molecular chaperone systems are characterized by the binding of exposed hydrophobic regions and are commonly coupled with an array of co-chaperones that aid in the regulation of their activity and specificity[3–5].

Members of the Hsp70 family are among the most ubiquitous molecular chaperones and are found in all kingdoms of life[6,7]. Hsp70s exhibit promiscuous binding of short hydrophobic sequences within client proteins and use ATPase activity to ensure that protein folding pathways, translocation events, protein degradation, and signaling cascades are maintained in both basal and stressed cellular environments[3–6,8]. Unsurprisingly, several diseases and cancers have been attributed to the dysfunction of Hsp70 activity[9–12]. Hsp70s also collaborate with two classes of co-chaperones, termed J-proteins (Hsp40s) and nucleotide exchange factors (NEFs), that stimulate ATP hydrolysis and release of ADP/substrate, respectively (Fig. 1A)[13–16]. These co-chaperones are critical for Hsp70 function as they endow substrate specificity and catalytic efficiency to an otherwise promiscuous and inefficient ATPase[5,13–16].

In the canonical Hsp70 substrate capture and release cycle, Hsp70 binds ATP at its N-terminal nucleotide-binding domain (NBD) and adopts a compacted conformation, with the C-terminal substrate binding domain (SBD) interacting substantially with the NBD (Fig. 1A, B). Upon binding of a protein substrate at the SBD, Hsp70 cooperates with a stimulatory J-protein to hydrolyze ATP and release inorganic phosphate, adopting an extended conformation with the ADP-bound NBD and substrate-bound SBD separated via the conserved interdomain linker (IDL)[16–19]. In bacteria, mitochondria, and chloroplasts, the GrpE-family of NEFs interacts with substrate-bound Hsp70 to facilitate ADP and protein substrate release[16,20–23]. ATP binding results in rapid dissociation of the NEF and enables the release of nucleotide and substrate, thus resetting Hsp70 to its ATP-bound, substrate-free state (Fig. 1A)[20,21,24].

Department of Chemistry and Biochemistry, University of California, San Diego, La Jolla, CA, USA. ✉e-mail: mherzik@ucsd.edu

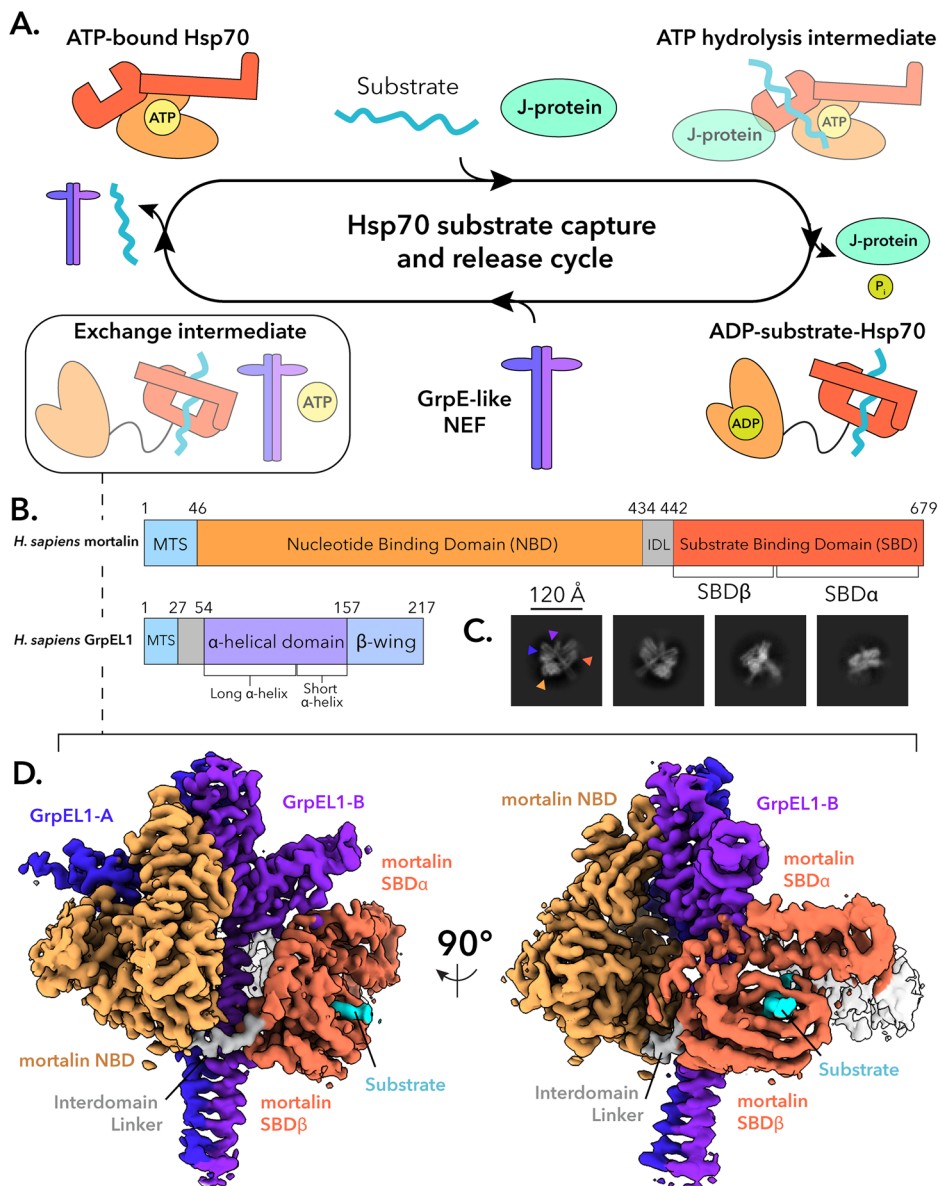

**Fig. 1 | Structural determination of the *Hs*mortalin_{R126W}-GrpEL1 complex.**
**A** Hsp70 proteins collaborate with co-chaperones to perform substrate capture and release. Mitochondrial Hsp70 (mortalin) interacts with a J-protein partner to capture substrates and hydrolyze ATP. Once substrate-bound, mortalin interacts with GrpE-like NEFs to facilitate substrate release via an unknown mechanism. NEF: nucleotide exchange factor. **B** Domain topology of *Hs*mortalin and *Hs*GrpEL1. MTS:

mitochondrial targeting sequence. IDL: interdomain linker. **C** Representative 2-D classes of the *Hs*mortalin_{R126W}-GrpEL1 complex. Colored arrows correspond to the densities observed in (**D**). **D** DeepEMhancer-sharpened map of the *Hs*mortalin_{R126W}-GrpEL1 complex colored by subunits and subdomains. The structure represents the exchange intermediate depicted in (**A**).

In mitochondria and chloroplasts, Hsp70s and their co-chaperones mediate not only protein folding and protein complex assembly but also are essential for the translocation of proteins across the membranes of these organelles[16,19,25,26]. Mammalian mitochondria harbor a distinctive Hsp70 homolog (mtHsp70) known as mortalin (*Hs*HSPA9 in humans)[27–29]. Mortalin and the human mitochondrial NEF, GrpEL1, together facilitate several key biological processes within the mitochondrial matrix (Fig. 1B)[30–33]. Mortalin and GrpEL1 work with two distinct J-protein complexes to perform their functions. The DNAJA3 complex mediates protein folding and protein complex assembly[34–36], while the Pam16/Pam18 complex aids the translocation of proteins through the translocase of the inner mitochondrial membrane-23 (TIM23) complex, via the presequence associated motor (PAM) complex[16,37–39].

Insights into mortalin–GrpEL1 function have come primarily from studies of their bacterial counterparts, DnaK and GrpE[21,24,40,41]. While these studies have revealed a putative mechanism for GrpE-mediated nucleotide release, existing structures of bacterial proteins are limited by the use of truncated constructs or flexible and disordered regions[22,40,42]. These limitations have not only precluded our understanding of critical substrate binding and release allosteric mechanisms, but also have hindered the elucidation of these mechanisms in eukaryotic contexts. Thus, comprehensive structural and mechanistic studies of eukaryotic Hsp70s, in particular mortalin with its co-chaperones, are needed.

To elucidate the mechanisms guiding nucleotide and substrate exchange in mortalin, we performed cryoEM structural analyses of human mortalin–GrpEL1 complexes. Herein, we present structures of

full-length human mortalin in a nucleotide-free and substrate-bound state in complex with full-length GrpEL1. Comparisons with bacterial DnaK and GrpE structures reveal insights into the roles of GrpE-family NEFs in mediating nucleotide and substrate release. Finally, we combine cryoEM and molecular dynamics (MD) to characterize an interface between GrpEL1 and the mortalin SBD that likely plays a key role in substrate release. Overall, this work enables comparative analyses between bacterial DnaK-GrpE and human mortalin-GrpEL1 chaperone systems and points to a conserved mechanism for GrpE-family NEFs in nucleotide and substrate release from Hsp70-family chaperones.

## Results

### Structure of the human mortalin$_{R126W}$-GrpEL1 complex

To better understand how eukaryotic Hsp70 proteins dynamically interact with their co-chaperones and substrates, we reconstituted a complex of full-length human mortalin and GrpEL1 with a soluble chimeric construct of Pam16/Pam18 (see "Methods" and Supplementary Fig. 1). To stabilize the complex for structural analysis, we introduced a point mutation, R126W, into mortalin (mortalin$_{R126W}$) that is associated with EVEN-PLUS syndrome and has been shown to reduce mortalin's ATPase activity and conformational flexibility[12]. Attempts to form a mortalin–GrpEL1 complex with wild-type mortalin resulted in inconsistent complex formation, whereas mixing GrpEL1 with mortalin$_{R126W}$ resulted in reproducible complex. Incubation of these components and subsequent separation via size exclusion chromatography yielded a high molecular weight species that appeared to comprise mortalin$_{R126W}$, GrpEL1, and a lower molecular weight species (Supplementary Fig. 2). While Pam16/Pam18 is not bound to this complex, its inclusion in the protein mixture was nonetheless important to mediate proper assembly of the stable mortalin$_{R126W}$-GrpEL1 complex. We performed cryoEM analysis of the mortalin$_{R126W}$–GrpEL1 complex, which yielded two-dimensional (2-D) class averages representing the canonical GrpE-like extended dimer with apparent density for mortalin on both sides of the α-helical stalk domain (Fig. 1C). Initial reconstructions yielded a cryoEM density containing the mortalin NBD and GrpEL1 dimer, however, density for the substrate binding domain (SBD) was weak. Heterogeneous refinement using reference volumes with and without SBD density allowed for the separation of particles that contained the entirety of the SBD (Supplementary Fig. 3). Further refinement yielded a 2.96 Å resolution cryoEM map describing a complex consisting of full-length mortalin$_{R126W}$ bound to a protein substrate and a full-length GrpEL1 homodimer (Fig. 1D and Supplementary Fig. 4).

This cryoEM structure depicts a 1:2 stoichiometry of mortalin$_{R126W}$:GrpEL1 with mortalin$_{R126W}$ in an elongated state stabilized by numerous interactions across both protomers of the GrpEL1 dimer (Fig. 1D). Firstly, we observe an expanded NBD devoid of nucleotide that is mediated by extensive contacts with one protomer of GrpEL1, termed GrpEL1-A. Secondly, we detail the significant bending of the GrpEL1 α-helical stalk domain that enables interactions between the GrpEL1 stalk and mortalin$_{R126W}$ NBD and SBD. Finally, this structure provides insights into the interactions between the SBDα helical lid of mortalin and the second GrpEL1 protomer, GrpEL1-B. Critically, the lack of EM density within the nucleotide-binding pocket but strong density within the substrate-binding pocket suggests that our structure represents a previously unobserved intermediate state and suggests a step-wise mechanism for the nucleotide and substrate release action of GrpEL1. Furthermore, since several bacterial DnaK-GrpE structures have been published to date[22,40,42], this structure enables novel comparisons between the bacterial and human systems, detailed below.

### GrpEL1 facilitates full expansion of the mortalin NBD for nucleotide release

In our mortalin$_{R126W}$–GrpEL1 structure, the NBD of mortalin$_{R126W}$ interacts with GrpEL1-A at two distinct interfaces. At the first interface,

subdomain IB of the NBD contacts the β-wing domain of GrpEL1-A via a salt bridge between R107 in mortalin$_{R126W}$ and the carbonyl group of P172 in GrpEL1-A (Fig. 2A). Additional van der Waals interactions between V110 in mortalin$_{R126W}$ and A177 in GrpEL1-A provide additional stabilization to this interface, resulting in 108 Å$^2$ of buried surface area. The bulk of the NBD-GrpEL1 interactions, however, are located between the short α-helices of GrpEL1-A and GrpEL1-B. At the second interaction interface, several electrostatic and van der Waals interactions stabilize the IIB lobe of mortalin's NBD against GrpEL1-A (Fig. 2A), yielding 219 Å$^2$ of buried surface area. Notably, these interfaces are present in bacterial DnaK-GrpE structures and are thought to contribute to the opening of the NBD for nucleotide release[22,40,42].

Several structures of the bacterial DnaK NBD complexed with GrpE have suggested a nucleotide release mechanism whereby the IIB-NBD lobe rotates outward through interactions with the β-wing domain of GrpE, facilitating ADP release[22,40,42]. We observe a similar expansion of the mortalin NBD in our structure, as well as a lack of EM density within the nucleotide-binding pocket (Fig. 2A and Supplementary Fig. 5). To probe the potential differences between the human and bacterial nucleotide release mechanisms, we performed structural comparisons between various DnaK-GrpE structures and our mortalin$_{R126W}$–GrpEL1 structure. Aligning these structures to the NBD IA and IB lobes allowed for visualization of the motions associated with GrpEL1-mediated rotations of the IIB lobe. Using the inward-facing α-helix within the IIB lobe as a point of reference, our structure exhibits ~15° of rotation compared to the ADP-PO$_4$-bound $Mg$DnaK structure (PDB: 5OBW)[43] and ~6° of increased rotation compared to the $Mt$DnaK-GrpE structure (PDB: 8BG3)[40] (Fig. 2B). Compared to existing mortalin NBD structures (PDB: 4KBO, 6NHK)[12,31], we observe a similar widening of the IIB lobe (Supplementary Fig. 6). Together, these analyses indicate conserved mechanisms of GrpE(L1)-mediated ADP release from the (mt)Hsp70 NBD across kingdoms.

Several conserved residues in the Hsp70 nucleotide binding pocket have been identified to stabilize bound nucleotide[6,44–46]. In our structure, nucleotide-interacting residues residing in the IA-NBD lobe are in similar positions relative to the ATP-bound state. However, residues K316 and S320 (K250 and S254 in $Mg$DnaK)[43] are displaced, abolishing stabilizing interactions of the ribose and adenosine rings of a bound nucleotide (Fig. 2C–E). This suggests that displacement of key residues in the IIB lobe destabilizes the bound nucleotide and contributes to ADP release.

### The mortalin interdomain linker facilitates bending of the GrpEL1 stalk

In our mortalin$_{R126W}$–GrpEL1 structure, we observe EM density for the interdomain linker (IDL) region that spans the nucleotide and substrate binding domains. Here, the mortalin$_{R126W}$ IDL is proximal to the GrpEL1 α-helical dimer with residues V435 and L438 within the hydrophobic IDL interacting with L82 in GrpEL1-B (Fig. 3A). Electrostatic and hydrogen bonding interactions between D434 in the IDL and K79 and Y78 in GrpEL1-B further stabilize the mortalin$_{R126W}$ IDL (Fig. 3A), contributing 146 Å$^2$ of surface area. Previous structures of bacterial DnaK bound to GrpE have described a similar IDL-GrpE interaction[31,40,42]. To further evaluate the significance of this interaction, we compared the residues of this interacting region between human and bacterial Hsp70-GrpE systems. Interestingly, while the IDL region is highly conserved across human and bacterial sequences, the corresponding interacting region within GrpE-like species varies significantly (Fig. 3B).

Interaction between the mortalin$_{R126W}$ IDL and GrpEL1 appears to be facilitated by significant bending of the GrpEL1 α-helical domain. Previous structures of bacterial GrpE and DnaK-GrpE that contain the IDL of DnaK describe a similar bending of the GrpEL1 stalk but to varying magnitudes[31,40,42]. To quantify the degree of GrpEL1 bending, we compared our structure and other DnaK-GrpE structures to the

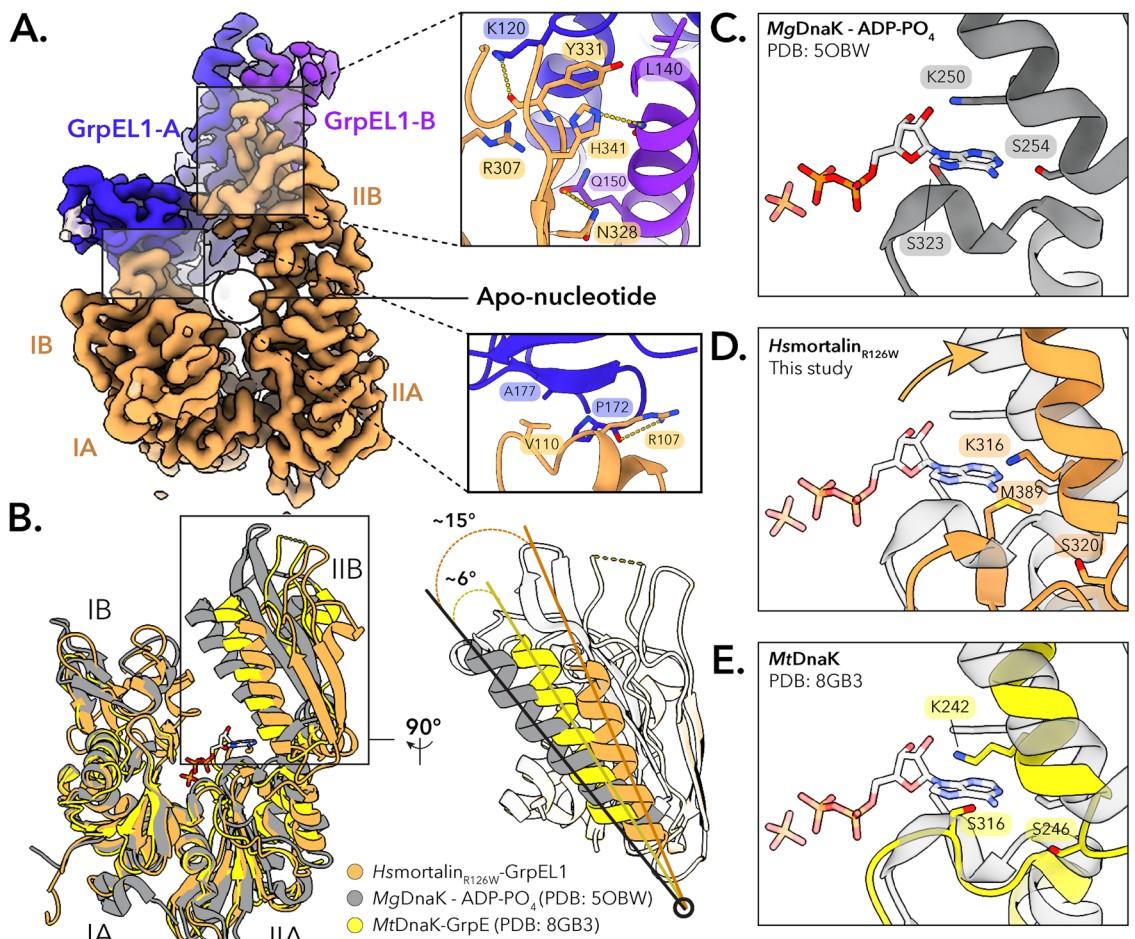

**Fig. 2 | The *Hs*mortalin_{R126W} NBD is fully expanded upon interaction with GrpEL1. A** Interface mapping between the mortalin_{R126W} IIB-NBD lobe and the GrpEL1-B short α-helix, and the mortalin_{R126W} IB-NBD lobe and the GrpEL1-A β-wing domain. The lack of EM density within the NBD suggests an apo-nucleotide NBD. **B** Structural comparison between the NBDs of *Hs*mortalin_{R126W}-GrpEL1, *Mt*DnaK-GrpE (PDB: 8GB3), and ADP-PO_4-bound *Mg*DnaK (PDB: 5OBW). Compared to ADP-PO_4-bound *Mg*DnaK, the IIB-NBD lobe of Hsmortalin_{R126W} expands ~15° upon interaction with GrpEL1. **C–E** Comparison of the ATP binding residues in the IIB-NBD lobe. ATP-stabilizing interactions in the IIB-NBD lobe of Hsmortalin_{R126W}-GrpEL1 are removed by the movement of the IIB-NBD lobe. ADP-PO_4 is modeled from *Mg*DnaK-ADP-PO_4. The semi-transparent model in panels (**D** and **E**) represents the *Mg*DnaK-ADP-PO_4 structure (PDB: 5OBW).

AlphaFold2[47] prediction of a human GrpEL1 dimer, which describes a linear α-helical domain. Compared to the predicted GrpEL1 structure, our mortalin_{R126W}-GrpEL1 structure exhibits a ~13° bending of the GrpEL1 α-helical domain. In contrast, the structure of *Mt*DnaK-GrpE[40] exhibits a ~26° bending of the α-helical domain (Fig. 3C).

## Substrate interactions within the mortalin_{R126W}–GrpEL1 complex

Our EM density reveals the entirety of the mortalin substrate binding domain, with the SBD α-helical subdomain resting on top of the β-sandwich subdomain (Fig. 4A). We observed strong density for a peptide bound to the SBD substrate binding pocket, indicating that our purified complex is substrate-bound. Behind this clear substrate density, we observed weaker density that matches the mortalin SBD β-sandwich subdomain (Fig. 4B). Hsp70s have been observed to bind themselves as substrates as in the *Gk*DnaK-GrpE (PDB: 4ANI)[42] crystal complex. Thus, we hypothesized that the mortalin_{R126W} protomer in complex with GrpEL1 was bound to the interdomain linker region of another mortalin protomer. Notably, we did not observe density for the NBD of a second mortalin protomer. Given the stability of the NBD, it is unlikely that the absence of NBD density can simply be attributed to signal averaging. Rather, we propose that the observed bound substrate is a mortalin truncation product containing only the IDL and SBD. We performed size exclusion chromatography coupled with

multi-angle light scattering (SEC-MALS) on our mortalin_{R126W}-GrpEL1 complex, which indicated the presence of 152.5 kDa species (Supplementary Fig. 2). This molecular weight agrees with a complex consisting of one full-length mortalin (70.2 kDa), two GrpEL1 protomers (25.5 kDa each) and an IDL-SBD mortalin truncation product (28.5 kDa). With these factors in mind, combined with Hsp70's propensity to bind hydrophobic peptides[48,49], we modeled the bound peptide as VLLLDVT, corresponding to residues 435–441 of mortalin's IDL region, into our cryoEM density (Fig. 4C). Moreover, docking of the mortalin NBD connected to an IDL bound to the substrate binding pocket appears to be capable of accommodating a full-length mortalin (Supplementary Fig. 7). Taken together, these observations suggest that our mortalin_{R126W}-GrpEL1 complex is bound to a second partial mortalin protomer as a substrate.

Hsp70 substrate specificity and binding modes have been studied extensively using various biochemical and structural techniques[5,6,48,49]. Current Hsp70 binding models are based upon DnaK-substrate interactions and describe a hydrophobic binding pocket that can accommodate five to seven hydrophobic residues[5,6,48,49]. The central residue, termed the 0th position, favors the binding of leucine residues, whereas the flanking residues are typically occupied by hydrophobic residues[5,49]. In our model we identify a mortalin substrate binding mode consistent with the canonical DnaK substrate binding model. Here, a leucine is oriented in the 0th position and makes hydrophobic

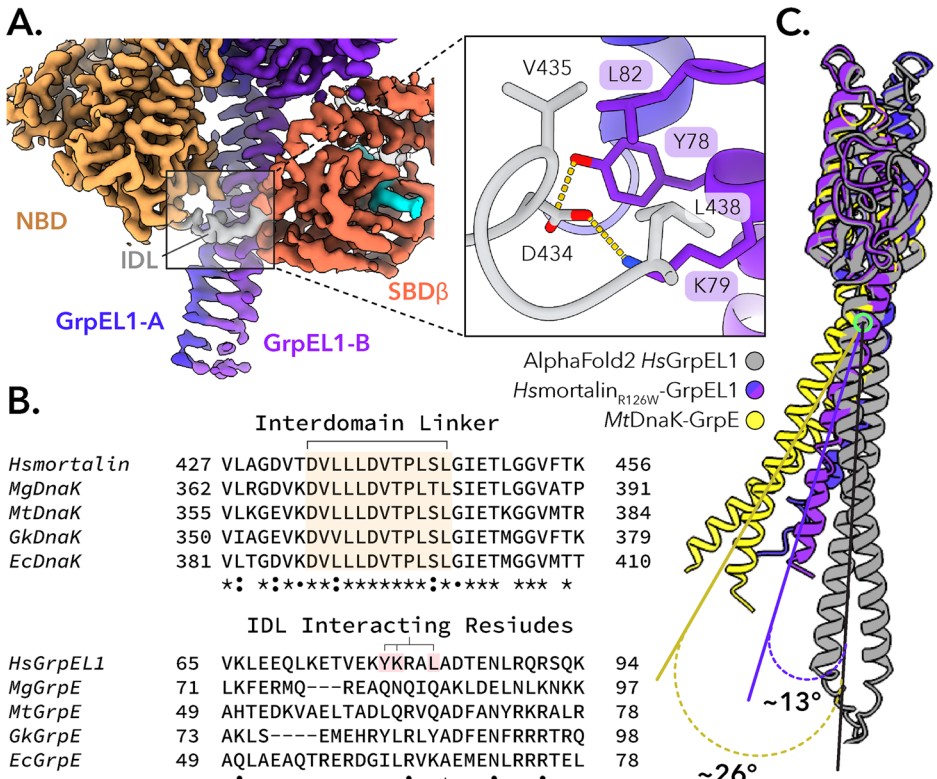

**Fig. 3 | The mortalin interdomain linker is stabilized by interaction with the GrpEL1 long α-helix. A** Interaction mapping between the $Hs$mortalin$_{R126W}$ linker and the GrpEL1 long α-helix. **B** Multiple sequence alignment between Hsp70 and GrpE-like species. The interdomain linker of Hsp70 is highly conserved, whereas the interacting GrpE-like region is variable. $Hs$: homo sapiens, $Mg$: mycoplasma genitalium, $Mt$: mycobacterium tuberculosis, $Gk$: geobacillus kaustophilus, $Ec$: escherichia coli. **C** Superposition of GrpE-like species from our $Hs$mortalin$_{R126W}$-GrpEL1 structure, $Mt$DnaK-GrpE (PDB: 8GB3), and the AlphaFold2[47] prediction of $Hs$GrpEL1.

contacts with I484, and F472. Adjacent to the 0th leucine are additional flanking leucine residues which make additional hydrophobic contacts with L450, A475, and V482 (Fig. 4C). Comparison between the mortalin$_{R126W}$ substrate and an $Ec$DnaK substrate (PDB: 4EZW)[50] displays a highly conserved binding mode with several central leucines forming the core of the peptide substrate (Fig. 4D). Overall, our structure suggests that mortalin substrate binding occurs, as expected, in a near identical mode compared to DnaK substrate binding.

**Interactions between GrpEL1-B and the SBD domain of mortalin$_{R126W}$**

In our structure, the GrpEL1-B protomer interacts with both of mortalin$_{R126W}$'s SBD subdomains (Fig. 5A). First, downstream of mortalin$_{R126W}$'s interdomain linker region several electrostatic and hydrogen bonding interactions between the SBDβ cleft and the N-terminal stalk of GrpEL1-B result in 180 Å² of buried surface area (Fig. 5B). Second, we observe an interaction interface between the SBDα helical domain and the β-wing domain of GrpEL1-B. Here, Y173 from GrpEL1-B inserts itself into the mortalin$_{R126W}$ SBDα, forming electrostatic and cation-pi stacking interactions with R578 and R574, respectively. In addition, D171 in GrpEL1-B forms an electrostatic interaction with R574 to form an interface with a total buried surface area of 159 Å² (Fig. 5C). To interrogate the prevalence of this interface, we performed a co-variational analysis between Hsp70 and GrpE homologs of residues at this interface.

First, we compared the electrostatic interaction between D171 in $Hs$GrpEL1 and R574 $Hs$mortalin across species. In all sequences compared, ranging from bacteria to higher eukaryotes, D171 ($Hs$GrpEL1) was highly conserved (Fig. 5D and Supplementary Fig. 8). Interestingly, R574 ($Hs$mortalin) corresponded to mostly arginines and lysines in other species which would allow preservation of the salt bridge spanning the GrpEL1-B β-wing and SBDα lid. In our structure, Y173 ($Hs$GrpEL1) appears buried within the SBDα lid and participates in electrostatic, cation-pi, and hydrophobic interactions with mortalin. While this tyrosine is highly conserved in vertebrates, this residue is substituted for mostly Asn in bacteria and lower eukaryotes. Interestingly, residues corresponding to R578 ($Hs$mortalin) are substituted for negatively charged (Asp/Glu) and hydrophobic (Leu) residues. Negatively charged residues would be capable of interacting with the substituted Asn and maintaining this GrpE β-wing-SBDα interface (Supplementary Fig. 9).

The interactions we observe between the GrpEL1-B β-wing and SBDα are exclusive to this interface since Y173 in GrpEL1-A does not appear to contact the NBD (Supplementary Fig. 10). Considering the unique interaction interfaces we observe between NBD-GrpEL1-A and SBD-GrpEL1-B, we designate unique identifiers to the GrpEL1 faces that interact with the NBD, termed β-wing face-N, or interact with the SBD, termed β-wing face-S (Fig. 5E). These unique interaction faces rationalize the necessity of a dimeric GrpEL1 and support the specific, asymmetric mortalin conformation with GrpEL1. Intrigued by the GrpEL1-B-SBDα interface, we hypothesized that interactions here contribute to GrpEL1-mediated substrate release. Thus, we mutated Y173 in GrpEL1 to an alanine (GrpEL1$_{Y173A}$) and performed cryoEM analysis on a mortalin$_{R126W}$-GrpEL1$_{Y173A}$ complex.

**CryoEM structure determination of mortalin$_{R126W}$-GrpEL1$_{Y173A}$**

Complex formation with mortalin$_{R126W}$ and GrpEL1$_{Y173A}$ yielded a similar profile on size exclusion chromatography compared to mortalin$_{R126W}$-GrpEL1$_{WT}$ (Supplementary Fig. 11). As previously described, the high molecular weight peak was concentrated and

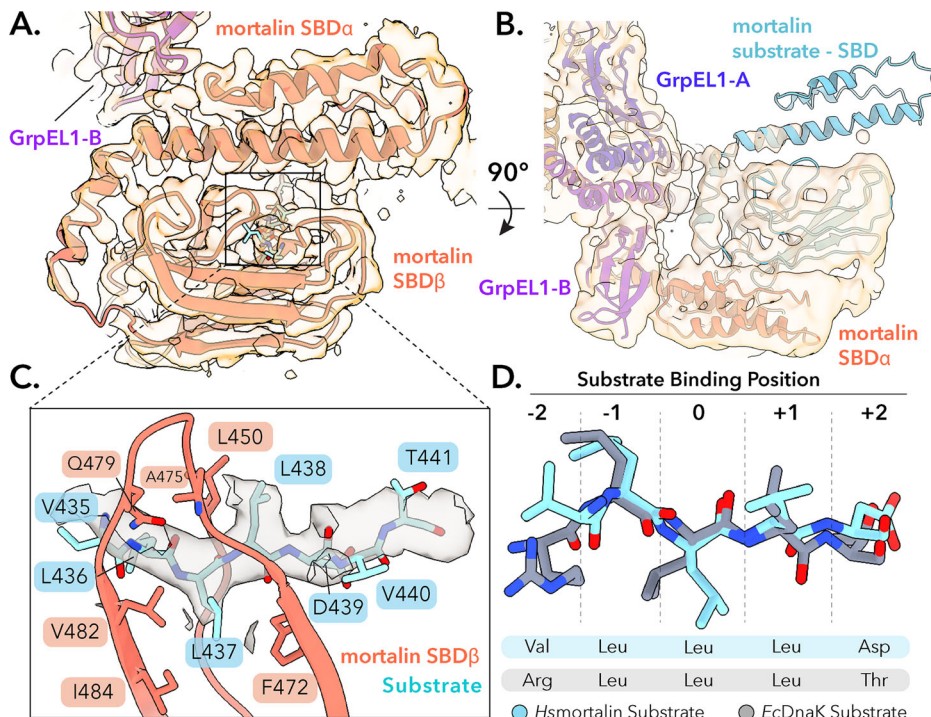

**Fig. 4 | *Hs*mortalin$_{R126W}$ in complex with GrpEL1 is substrate-bound to a mortalin truncation product. A** Model of the *Hs*mortalin$_{R126W}$ substrate binding domain (SBD) bound to a substrate fit into the DeepEMhancer-sharpened cryoEM map. **B** Rigid-body docking of the mortalin SBD into the posterior substrate EM density (low-pass filtered to 5 Å). **C** Residue mapping of the substrate within the mortalin substrate binding site. The density of the substrate is shown as a gray contour. **D** Comparison of the modeled *Hs*mortalin substrate with the crystal structure of *Ec*DnaK bound to the NRLLLTG peptide (PDB: 4EZW).

vitrified onto cryoEM grids. Our cryoEM analysis of mortalin$_{R126W}$-GrpEL1$_{Y173A}$ yielded two distinct populations differentiated by the presence or lack of density for the SBDα lid. We term these structures mortalin$_{R126W}$-GrpEL1$_{Y173A}$ and mortalin$_{R126W}$-GrpEL1$_{Y173A}$–lid (Supplementary Figs. 12, 13). Overall, these structures closely resemble our mortalin$_{R126W}$-GrpEL1$_{WT}$ structure and are similarly nucleotide-free. However, we observed notable differences in the IDL and SBD regions of our mortalin$_{R126W}$-GrpEL1$_{Y173A}$ structures. Firstly, our mortalin$_{R126W}$-GrpEL1$_{Y173A}$ structure harbors EM density for the entirety of the SBD, and similar to our mortalin$_{R126W}$-GrpEL1 structure, appears substrate bound to another mortalin$_{R126W}$ proteolysis product (Fig. 6A). Critically, we observe a large deviation of the SBD in our mortalin$_{R126W}$-GrpEL1$_{Y173A}$ structure compared to mortalin$_{R126W}$-GrpEL1$_{WT}$, with the SBD of mortalin$_{R126W}$-GrpEL1$_{Y173A}$ translated ~6 Å away from the GrpEL1-B β-wing (Fig. 6B). In our mortalin$_{R126W}$-GrpEL1$_{Y173A}$–lid structure, we discern diminished EM density for the IDL region and density only for the SBDβ subdomain (Fig. 6C). Importantly, we are unable to identify EM density of bound substrate that is observed in our other mortalin$_{R126W}$-GrpEL1 structures. Considering density for the SBDα subdomain is also absent in the mortalin$_{R126W}$-GrpEL1$_{Y173A}$–lid structure, we hypothesize that this complex harbors a disordered SBDα subdomain and may represent a post-substrate release state. A recently published structure of *Mt*DnaK-GrpE (PDB: 8GB3)[40] also harbors a disordered SBDα subdomain and is suggested to represent a post-substrate release conformation.

These findings are intriguing for several reasons. Firstly, positioning of the SBD in our mortalin$_{R126W}$-GrpEL1$_{Y173A}$ structure suggests that although Y173 in GrpEL1 is not necessary for mortalin$_{R126W}$-GrpEL1 complex formation, it is key in mediating interaction between the β-wing domain of GrpEL1-B and SBDα lid of mortalin. Secondly, the presence of the full SBD in mortalin$_{R126W}$-GrpEL1$_{Y173A}$ suggests that interactions between the GrpEL1-B stalk and SBDβ subdomain of mortalin are sufficient in stabilizing the entirety of the SBD. Finally, the lack of SBDα density in our mortalin$_{R126W}$-GrpEL1$_{Y173A}$–lid structure

suggests flexibility of the SBDα lid, especially in the absence of substrate.

## The SBDα lid and GrpEL1-B β-wing domains exhibit moderate flexibility

To investigate the flexibility in our structures, we performed anisotropic network modeling (ANM) of mortalin$_{R126W}$-GrpEL1$_{WT}$ and mortalin$_{R126W}$-GrpEL1$_{Y173A}$ (Supplementary Videos 1 & 2). Mode 1 of both mortalin$_{R126W}$-GrpEL1$_{WT}$ and mortalin$_{R126W}$-GrpEL1$_{Y173A}$ describe a motion in the SBDα subdomain that is perpendicular to the plane formed by SBDα and SBDβ that we term a lateral motion (Fig. 7A). Interestingly, we observe a secondary motion, most clearly observed in mode 2 in mortalin$_{R126W}$-GrpEL1$_{Y173A}$, that describes a medial motion whereby the SBDα moves parallel to the SBDα/β plane (Fig. 7B).

To validate the motions observed in our ANM analyses, we conducted all-atom molecular dynamics (MD) simulations on the mortalin$_{R126W}$-GrpEL1$_{WT}$ (Supplementary Videos 3–5) and mortalin$_{R126W}$-GrpEL1$_{Y173A}$ (Supplementary Videos 6–8) structures (three replicates of 150 nanoseconds each for a total of 450 ns). To assess the amount of lateral movement in these simulations, we defined vectors $v_1$ (between Ala569 and Glu597 in mortalin), $v_2$ (between Ala569 in mortalin and Lys200 in GrpEL1-B), and $v_3$ (between Ala569 in mortalin and Glu98 in GrpEL1-B) (Supplementary Fig. 14). The change in angle between vectors $v_1$ and $v_3$ were used to evaluate lateral motions whereas the change in angle between $v_1$ and $v_2$ was used to evaluate medial motions. Across all our analyses (Supplementary Figs. 15, 16), we observe both lateral and medial motions in mortalin$_{R126W}$-GrpEL1$_{WT}$ and mortalin$_{R126W}$-GrpEL1$_{Y173A}$. In our analysis of lateral SBD motions, we found that in both mortalin$_{R126W}$-GrpEL1$_{WT}$ and mortalin$_{R126W}$-GrpEL1$_{Y173A}$, the SBDα lid tended to reside at a substate 10–20° away from its starting position, around 95–110° as defined by vectors $v_1$ and $v_3$ (Fig. 7C and Supplementary Figs. 14–16). This is consistent with the motions observed in our ANM analysis. Moreover, comparisons between the SBD in mortalin$_{R126W}$-GrpEL1$_{WT}$

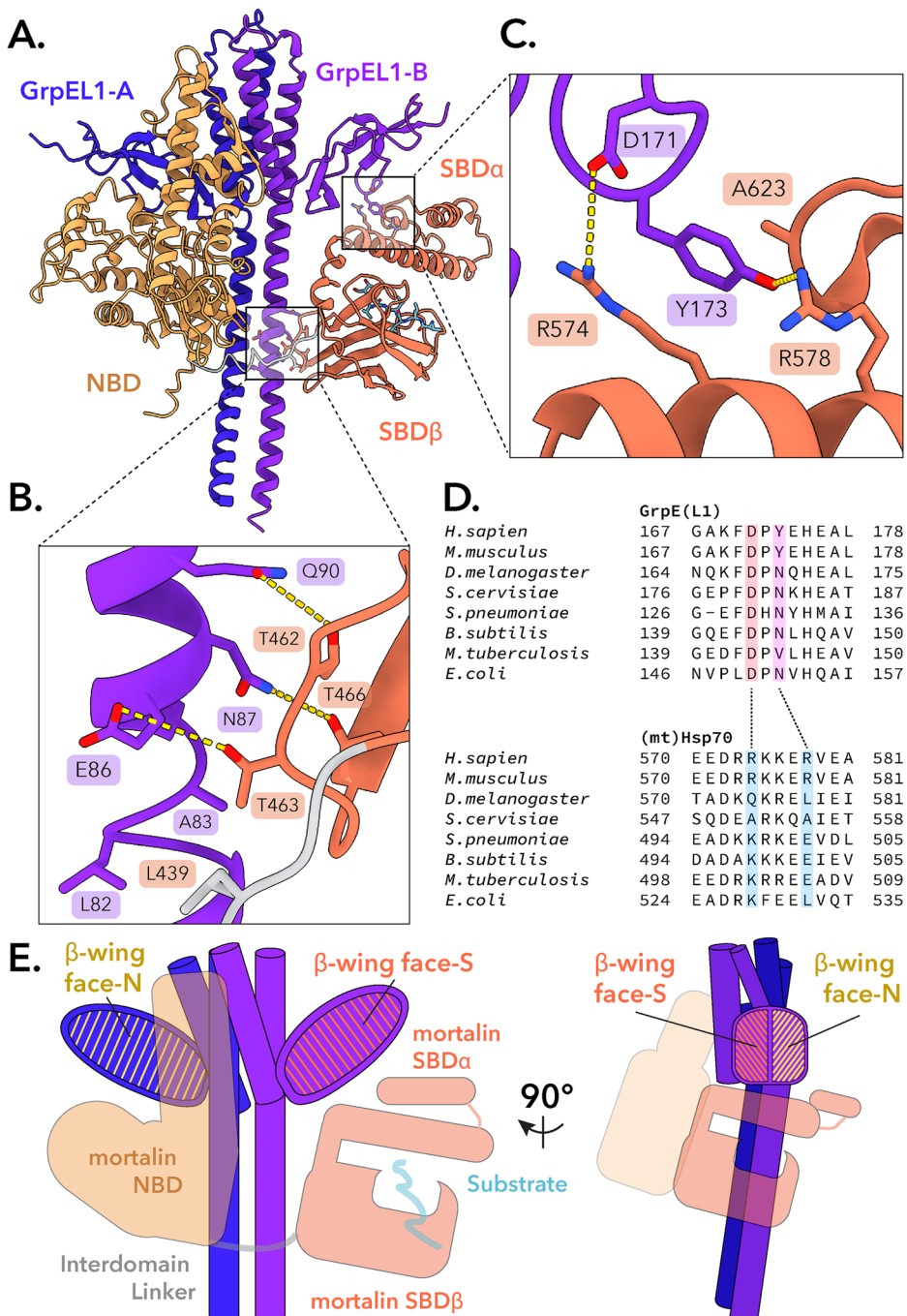

**Fig. 5 | GrpEL1 forms unique interactions with the *Hs*mortalin$_{R126W}$ NBD and SBD. A** Structural mapping of the *Hs*mortalin$_{R126W}$ SBD with the GrpEL1-B long α-helix and β-wing domain. **B** Residue mapping of the interactions between the GrpEL1-B long α-helix and *Hs*mortalin$_{R126W}$ SBDβ cleft. **C** Residue mapping of the interactions between the GrpEL1-B β-wing domain and SBDα helical domain.

**D** Multiple sequence alignments of the GrpEL1-B β-wing-SBDα interacting regions across GrpE-like and Hsp70 species. **E** Designation of the GrpEL1 β-wing face that interacts with the *Hs*mortalin$_{R126W}$ NBD (β-wing face-N) and the face that interacts with the SBD (β-wing face-S).

and existing Hsp70-SBD structures reveal positional variations of the SBDα lid, providing experimental evidence of lateral flexibility in the SBDα subdomain (Supplementary Fig. 17).

In our analysis of medial motions, we found that the SBDα lid also tended to move 10–20° away from its starting position, either away from GrpEL1-B, in the case of mortalin$_{R126W}$-GrpEL1$_{WT}$, or towards GrpEL1-B for mortalin$_{R126W}$-GrpEL1$_{Y173A}$ (Fig. 7D and Supplementary Figs. 15, 16).

To assess motions within the individual domains of mortalin$_{R126W}$-GrpEL1$_{WT}$ and mortalin$_{R126W}$-GrpEL1$_{Y173A}$, we performed RMSD

analyses on the GrpEL1-B β-wing, the SBDα lid, and the SBDβ subdomains (Supplementary Figs. 18, 19). In both mortalin$_{R126W}$-GrpEL1$_{WT}$ and mortalin$_{R126W}$-GrpEL1$_{Y173A}$, the SBDα lid exhibited the most deviation with 7.8 ± 0.7 Å average deviation in the WT complex and 8.7 ± 1.1 Å average deviation in the Y173A complex (Supplementary Figs. 18, 19, Average RMSD). Notably, the RMSD of the GrpEL1-B β-wing was different between the WT and Y173A complex, with mortalin$_{R126W}$-GrpEL1$_{WT}$ exhibiting 3.7 ± 0.9 Å average deviation compared to the mortalin$_{R126W}$-GrpEL1$_{Y173A}$ GrpEL1-B β-wing exhibiting 5.5 ± 0.6 Å average deviation.

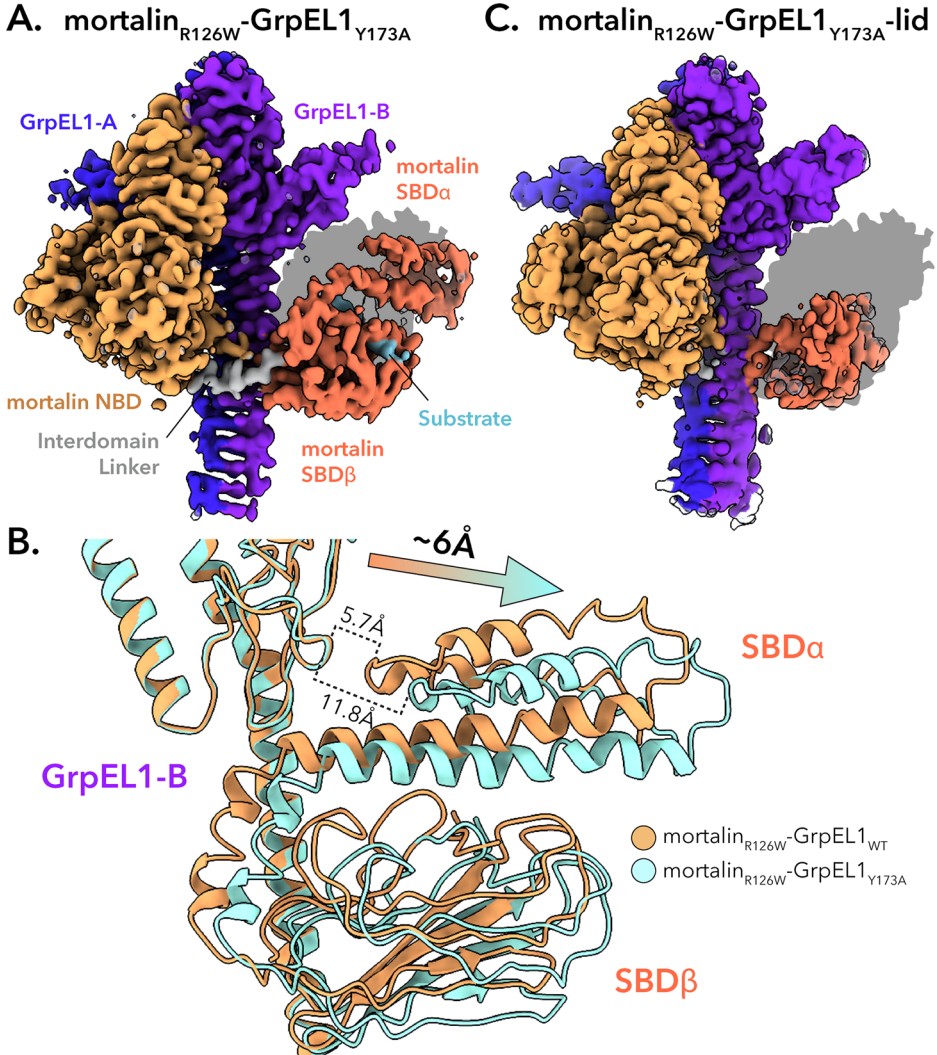

**Fig. 6 | Mutation of Y173A in GrpEL1 results in a shift of the *Hs*mortalin SBD.**
**A** DeepEMhancer-sharpened map of mortalin$_{R126W}$-GrpEL1$_{Y173A}$. Shadow represents the silhouette of the mortalin$_{R126W}$-GrpEL1$_{WT}$ SBD position. **B** Superposition of mortalin$_{R126W}$-GrpEL1$_{WT}$ and mortalin$_{R126W}$-GrpEL1$_{Y173A}$. In the mortalin$_{R126W}$- GrpEL1$_{Y173A}$ structure, the SBD is translated ~ 6 Å away from GrpEL1-B compared to WT. **C** DeepEMhancer-sharpened map of mortalin$_{R126W}$-GrpEL1$_{Y173A}$–lid. Shadow represents the silhouette of the mortalin$_{R126W}$-GrpEL1$_{WT}$ SBD position.

Taken together, our analyses indicate that the SBD-α lid and GrpEL1-B β-wing domains are moderately dynamic under the experimental conditions surveyed. We found that the SBDα lid tends to favor substates 10–20° away from its starting position in both the lateral and medial planes. We also observe that decoupling the SBD-α:GrpEL1-B interface via Y173A in GrpEL1 results in a change in dynamics only for the GrpEL1-B β-wing. Interestingly, on the timescale of our simulations, the SBD remained closed throughout. It's possible that the simulation timescale was not sufficient to capture an SBD open state. In addition, steps such as ATP binding may be required to initiate the complex for substrate release.

## Discussion

We used cryoEM to investigate how human GrpEL1 facilitates nucleotide exchange and substrate release of human mortalin. Our structure describes a 1:2 stoichiometry between mortalin and GrpEL1 that is analogous to previously observed bacterial DnaK-GrpE structures[22,40,42]. Importantly, we do not observe EM density for nucleotide in the NBD but strong density within the SBD, suggesting GrpEL1 facilitates nucleotide exchange prior to substrate release. We further identify unique interfaces between mortalin-GrpEL1 and a key interaction stabilizing a distinct mortalin SBD

conformation. Together, our analyses corroborate over two decades of work on Hsp70-GrpE complexes and enable us to speculate on the mechanisms underlying GrpEL1-mediated substrate release.

In our cryoEM structure of full-length *Hs*mortalin$_{R126W}$-GrpEL1, we observe a significant separation of the NBD lobes mediated by GrpEL1-A's β-wing domain. Compared to bacterial DnaK-GrpE structures[22,40,42], our complex shows a larger rotation in the IIB-NBD lobe. Considering the high sequence similarity between mortalin-GrpEL1 and DnaK-GrpE, it is unlikely that this difference is solely attributed to human vs. bacterial systems. Instead, we propose that docking of the SBD to the GrpEL1-B β-wing domain enables full expansion of the NBD, minimizing the chances of premature substrate release. In addition, we note a moderate (~ 13°) bending of the GrpEL1 long α-helical stalk that is likely constrained by myriad interactions across mortalin-GrpEL1. Finally, our identification of β-wing face-N and β-wing face-S underscores the importance of an asymmetric GrpEL1 homodimer and rationalizes the 1:2 stoichiometry of Hsp70:GrpE-like structures.

The interaction between GrpEL1-B and the SBD in our mortalin$_{R126W}$-GrpEL1 complex provides new insights that were lacking in previous DnaK-GrpE homolog structures. We observe clear EM density spanning the GrpEL1-B β-wing domain and SBDα lid subdomain, indicative of a stable interface. Mutational and structural

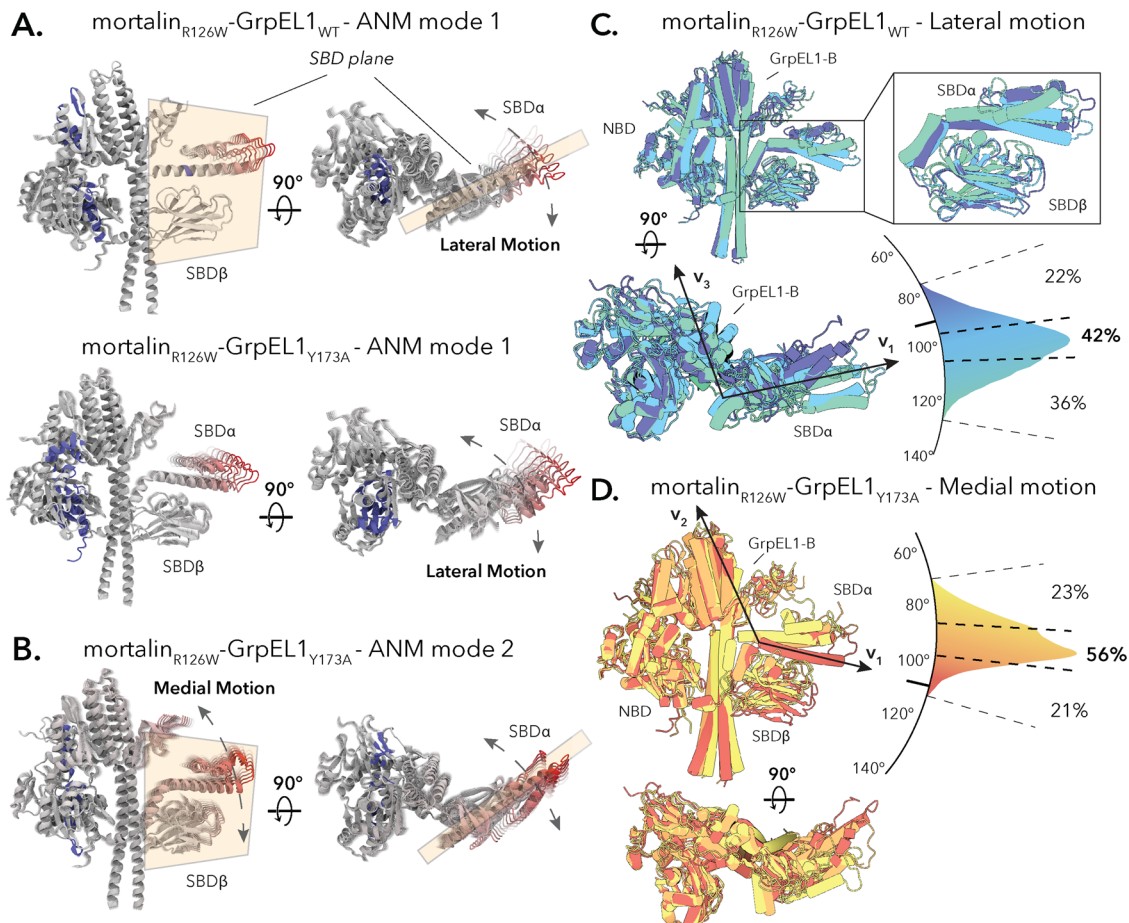

**Fig. 7 | Flexibility analysis of mortalin_R126W-GrpEL1_WT and mortalin_R126W-GrpEL1_Y173A. A** Anisotropic network modeling (ANM) mode 1 of mortalin_R126W-GrpEL1_WT and mortalin_R126W-GrpEL1_Y173A describes a lateral motion of the SBD. The SBD plane is colored in beige. **B** ANM mode 2 of mortalin_R126W-GrpEL1_Y173A describes a medial motion of the SBD. The SBD plane is colored in beige. Regions colored in blue describe low amounts of motion whereas regions in red describe high amounts of motion. **C** Analysis of lateral motion in replicate 1 of the mortalin_R126W-GrpEL1_WT all-atom simulation. The angular plot denotes the angle between vectors $v_1$ and $v_3$ (Supplementary Fig. 14) throughout a 150 ns replicate. The structural snapshot in dark blue represents a representative substate from 78 to 95.9°, in light blue a representative substate from 96 to 105°, and in seafoam green a representative substate from 105.1 to 134°. The percentage of trajectory in each region is annotated on the angular plot. The angle of the initial substate is denoted by a black line on the angular plot. **D** Analysis of medial motion in replicate 1 of the mortalin_R126W-GrpEL1_Y173A all-atom simulation. The angular plot denotes the angle between vectors $v_1$ and $v_2$ (Supplementary Fig. 14) throughout a 150 ns replicate. The structural snapshot in yellow represents a representative substate from 74 to 86.9°, in orange a representative substate from 87 to 96°, and in red a representative substate from 96.1 to 114°. The percentage of trajectory in each region is annotated on the angular plot. The angle of the initial substate is denoted by a black line on the angular plot.

analyses of the mortalin_R126W-GrpEL1_Y173A complex highlight the crucial role of Y173 in maintaining connectivity at this interface. However, the presence of an intact, substrate-bound SBD in our mortalin_R126W-GrpEL1_Y173A structure suggests that interactions between SBDβ and GrpEL1-B stalk are sufficient to maintain the closed SBD conformation proximal to GrpEL1.

In our study, we used a mutant construct of mortalin harboring the R126W point mutation. This mutation has been implicated in EVEN-PLUS syndrome, and exhibits decreased ATPase activity and stability[12]. In mortalin_R126W-GrpEL1_WT we do not observe R126W making additional contacts within the NBD (Supplementary Fig. 20A). In the *Mt*Dnak-GrpE structure (PDB: 8GB3)[40], the corresponding S75 in *Mycobacterium tuberculosis* does not appear to contribute to complex stabilization, supporting the idea that R126 does not contribute to mortalin_R126W-GrpEL1 complex formation (Supplementary Fig. 20B). Interestingly, upon examination of the AlphaFold2[47] predicted structure of human mortalin, R126 appears to form contacts with T271 in NBD lobe II (Supplementary Fig. 20C). Thus, it is possible that R126 is important for stabilization of a closed NBD amenable for ATP binding.

Our analyses provide insights into the architecture and dynamics of *Hs*mortalin-GrpEL1 and enable us to propose a new mechanism of GrpEL1-mediated regulation of mortalin. Here, ADP-bound mortalin with substrate encounters GrpEL1 and adopts an asymmetric conformation with the NBD interacting with β-wing face-N of GrpEL1-A, and SBDα interacting with β-wing face-S of GrpEL1-B. GrpEL1-A interactions with the IIB-NBD lobe result in expansion of the NBD and release of ADP. The SBDα lid is moderately flexible, potentially priming the complex for subsequent substrate release. Following ATP binding to the NBD we propose two potential pathways towards substrate release and complex dissociation. In the step-wise pathway, the NBD dissociates from GrpEL1 first and pulls away from SBDβ. The SBDα lid remains in contact with GrpEL1-B as SBDβ opens and enables substrate release, followed by full complex dissociation. In an alternative pathway, dissociation of the NBD and SBD are concerted, and the substrate is released by flexibility of the SBD upon decoupling of the SBDα lid-GrpEL1-B interface (Fig. 8). Further studies of the allosteric changes induced by ATP addition will be required to identify a comprehensive substrate release mechanism.

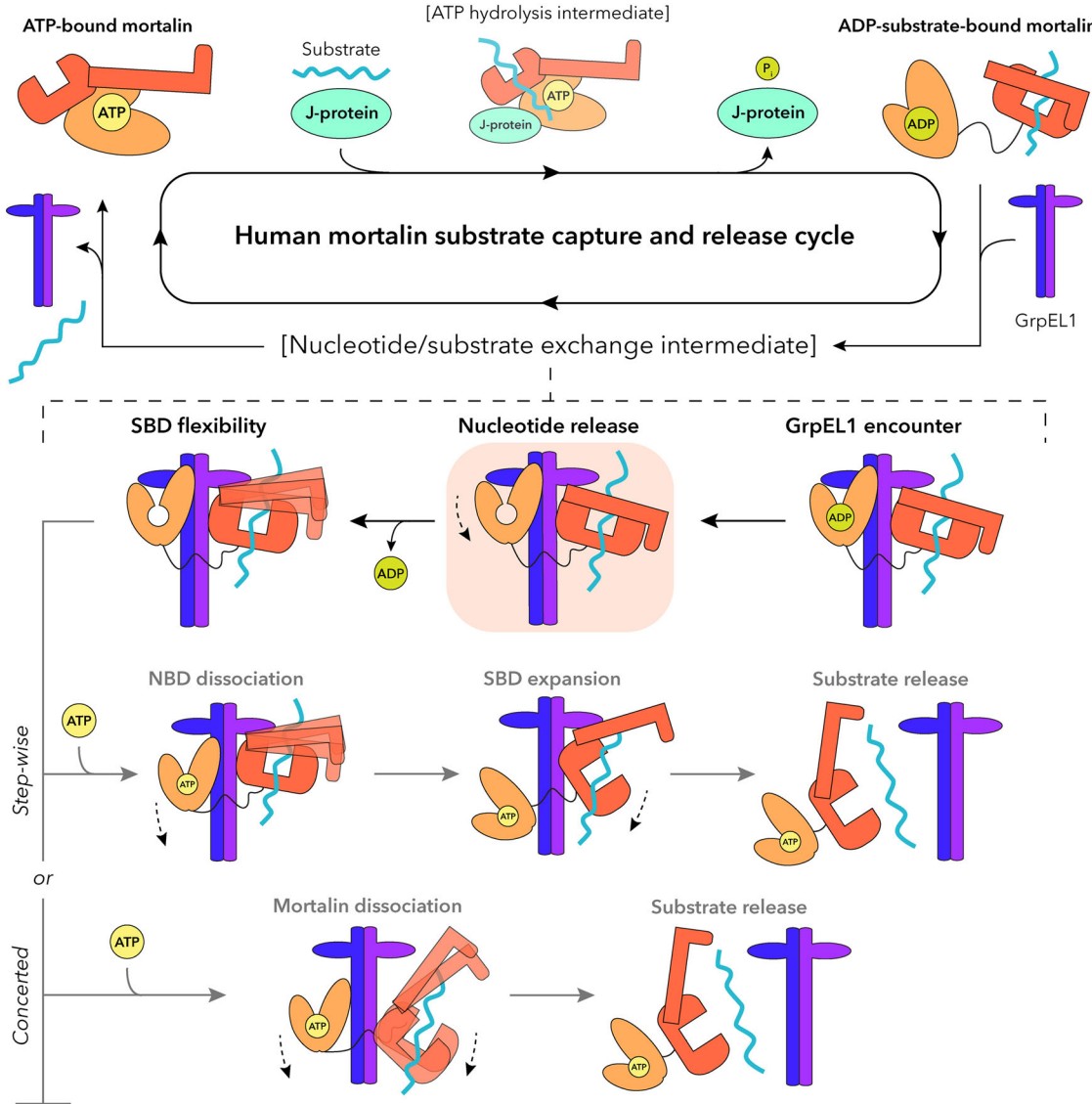

**Fig. 8 | GrpEL1-mediated nucleotide and substrate release mechanism of human mortalin.** Upon interaction with a substrate and J-protein, mortalin hydrolyzes ATP and stably interacts with substrate. GrpEL1 interacts asymmetrically with mortalin and facilitates ADP release via interactions at the NBD. Flexibility within the SBD loosens the SBDα subdomain. Following ATP binding in a step-wise mechanism, the NBD dissociates first, which enables the opening of the SBD and subsequent substrate release. Alternatively, ATP binding induces concerted dissociation of mortalin whereby decoupling of SBDα and GrpEL1-B enables substrate release. The intermediate representing the mortalin$_{R126W}$-GrpEL1$_{WT}$ structure is highlighted in orange.

Several biochemical studies have demonstrated the importance of the disordered N-terminal region of GrpE-like proteins in facilitating substrate release[21,40,51]. However, it should be noted that these studies have been limited to bacterial DnaK-GrpE systems, and given the substantial sequence divergence of the GrpEL1 N-terminus, partly due to the presence of a mitochondrial targeting sequence, we cannot assume conservation of this mechanism in eukaryotes. Nevertheless, we can speculate on the functional implications of eukaryotic GrpEL1 isoforms. Vertebrates possess two mitochondrially targeted GrpE-like isoforms, GrpEL1 and GrpEL2, with GrpEL2 non-essential but upregulated under oxidative stress[52–54]. Notably, the substitution of Y173 with H176 in GrpEL2 may suggest an additional level of regulation in stress-responsive exchange factors (Supplementary Fig. 21).

Together, our work reveals a near-complete molecular mechanism for GrpEL1 regulation of human mortalin. Additional investigations into how ATP mechanistically induces substrate release and complex dissociation will be key to obtaining a comprehensive understanding of this process. Importantly, our cryoEM structures of full-length human mortalin and GrpEL1 shed light on fundamental Hsp70 mechanisms ubiquitous to biology. These insights will be key in future studies to understand cellular responses in basal and stressed environments, and in the development of pharmaceuticals targeting Hsp70-related diseases.

## Methods

### Plasmid/construct design

Plasmids encoding full-length genes for human Hep1 and mortalin (C-terminal hexahistidine tag) in a pETDuet-1 vector, human GrpEL1 (C-terminal tandem Strep-Tag II and hexahistidine tags) in a pRSFDuet-1 vector, as well as human Pam16 and Pam18 in a pCDFDuet-1 vector were obtained from Bio-Basic. The mitochondrial targeting sequences for each gene (residues 1–49 for Hep1, 1–46 for mortalin, and 1–27 of GrpEL1) were subsequently removed prior to recombinant expression in *Escherichia coli*. The R126W mutant of mortalin (mortalin$_{R126W}$) and Y173A mutant of GrpEL1 (GrpEL1$_{Y173A}$) were generated using around-the-horn PCR. We designed a chimeric form of the peripheral

membrane proteins Pam16 and Pam18 to eliminate the transmembrane helices and improve the solubility of the J-like/J-protein heterodimeric Pam16/18 complex. Briefly, we fused the functional J-like (residues 56–121) and J-domains (residues 48–116) of *Hs*Pam16 and *Hs*Pam18, respectively, to the N- and C-terminus of *E. coli* maltose binding protein (MBP), respectively, with a 3x(GGS) linker between Pam16 and MBP and a GGS-HHHHHH-GGS linker between MBP and Pam18. We used AlphaFold2[47] to predict the structure of our Pam16(J-like)-MBP-H6-Pam18(J-domain) (Pam16/Pam18 chimera) chimeric complex and observed an architecture similar to that of the yeast Pam16/Pam18 crystal structure (PDB ID: 2GUZ), suggesting our chimeric complex could be functional (see Supplementary Fig. 1). All oligonucleotides were ordered from Integrated DNA Technologies and all plasmid sequences were confirmed by GENEWIZ. Verified plasmid sequences were transformed into the *E. coli* LOBSTR[55] strain for recombinant protein expression.

### Recombinant protein expression and purification

**mortalin.** A pETDuet-1 vector encoding for *Hs*Hep1 and *Hs*mortalin$_{R126W}$ was transformed into the *E. coli* LOBSTR[55] strain and grown in LB media containing 50 μg/mL ampicillin and 50 μM ZnCl$_2$ at 37 °C until the OD$_{600}$ reached 1.0. Protein expression was induced by the addition of 100 μM isopropyl β-D-1-thiogalactopyranoside (IPTG) and allowed to express for 4 hrs at 16 °C. Cells were centrifuged, collected, and resuspended in buffer A (20 mM Tris pH 8, 150 mM KCl, 50 μM ZnCl$_2$, 5 mM β-mercaptoethanol (BME), 25 mM imidazole, 1 mM MgCl$_2$, 1 mM CaCl$_2$, 1 μg/mL polyethyleneimine (PEI), 1 mM phenylmethylsulfonyl fluoride (PMSF), 5 mM benzamidine, 10 μM leupeptin, 1 μM pepstatin A, and 2 μg/mL aprotinin) with a small amount of DNase I (Millapore Sigma). Cells were lysed using a Branson SFX550 sonifier on ice. Lysed cells were centrifuged at 20,000 × *g* for 30 min at 4 °C, and the clarified supernatant was loaded onto a 5 mL HisTrap HP column (Cytiva) pre-equilibrated with buffer A at 1 mL/min using a peristaltic pump. The column was washed with buffer B (20 mM Tris pH 8, 150 mM KCl, 50 μM ZnCl$_2$, 5 mM BME, and 100 mM imidazole) until protein was no longer observed in the eluent, as visualized by mixing with Bradford reagent. The target protein was eluted using 5 column volumes (CVs) of buffer B containing 300 mM imidazole. Collected elutions were incubated for 30 min on ice with 0.5 mM AMP-PNP and 2 mM MgCl$_2$ prior to mixing with buffer C (20 mM Tris pH 8, 5 mM BME) to lower the [KCl] to 75 mM. A 5 mL HiTrap Q HP anion exchange chromatography column (Cytiva) was pre-equilibrated with buffer C supplemented with 75 mM KCl (Q buffer A). Mortalin$_{R126W}$ fractions were loaded onto the HiTrap Q HP column at 1 mL/min and washed with Q buffer A until the baseline was reached. A linear gradient was applied using Q buffer A to Q buffer B (20 mM Tris pH 8, 1 M KCl, 5 mM BME) from 0–100% Q buffer B over 20 CVs. Peak fractions containing mortalin$_{R126W}$ were pooled, concentrated, and applied to a Superdex 200 Increase 10/300 GL (Cytiva) that had been pre-equilibrated with buffer D (20 mM Tris pH 8, 150 mM KCl, 0.5 mM TCEP). Peak fractions were collected and analyzed using SDS-PAGE. The cleanest fractions were combined and concentrated to 1 mg/mL. Concentrated fractions were used immediately, or flash frozen for storage.

**GrpEL1$_{WT}$.** A pRSFDuet-1 vector encoding for *Hs*GrpEL1$_{WT}$ was transformed into the *E. coli* LOBSTR[55] strain and grown in LB media containing 25 μg/mL kanamycin at 37 °C until the OD$_{600}$ reached 1.0. Protein expression was induced by the addition of 200 μM IPTG and allowed to express for 3 hrs at 25 °C. Cells were centrifuged, collected, and resuspended in buffer A with DNase I and RNASe I (Millipore Sigma) added prior to sonication. Cells were lysed using a Branson SFX550 sonifier on ice. The supernatant was loaded onto 5 mL of Ni Sepharose High-Performance resin (Cytiva) that had been pre-

equilibrated with buffer A (without ZnCl$_2$) supplemented with 10 mM imidazole. The column was subsequently washed with buffer B containing 50 mM imidazole until protein was no longer observed, as detected by mixing with Bradford reagent. The target protein was eluted using 5 CVs of buffer B containing 300 mM imidazole. Elutions were pooled and loaded onto 3 mL of Strep-Tactin Superflow resin (IBA Lifesciences) that had been equilibrated with buffer E (20 mM Tris pH 8, 150 mM KCl, 5 mM BME) and incubated for 30 min at 4 °C. The column was washed with buffer E until no protein was observed in the eluent as detected by mixing with Bradford reagent. The target protein was eluted by the application of 5 CVs of buffer E containing 2.5 mM *d*-desthiobiotin (Sigma-Aldrich). Elutions were pooled, concentrated, and applied to a Superdex 200 Increase 10/300 GL (Cytiva) that had been pre-equilibrated with buffer D. Peak fractions were collected and analyzed using SDS-PAGE. The cleanest fractions were combined and concentrated to 0.8 mg/mL. Concentrated fractions were used immediately, or flash frozen for storage.

**GrpEL1$_{Y173A}$.** A pRSFDuet-1 vector encoding for the *Hs*GrpEL1$_{Y173A}$ mutant was transformed into the *E. coli* LOBSTR[55] strain. Recombinant protein expression and purification of GrpEL1$_{Y173A}$ were performed as described for GrpEL1$_{WT}$. Select peak fractions following SEC were visualized using SDS-PAGE. The cleanest fractions were combined and concentrated to 3.2 mg/mL. Concentrated fractions were used immediately, or flash frozen for storage.

**Pam16/18 chimera.** A pCDFDuet-1 vector encoding for the *Hs*Pam16/Pam18 chimera was transformed into *E. coli* LOBSTR[55] cells and grown in LB media containing 25 μg/mL of chloramphenicol and 50 μg/mL of spectinomycin at 37 °C until the OD$_{600}$ reached 0.6. Protein expression was induced by the addition of 100 μM IPTG and allowed to express for 16 hrs at 16 °C. Cells were centrifuged, collected, and resuspended in buffer A containing DNase I and RNAse I (Millipore Sigma). Cells were lysed using a Branson SFX550 sonifier on ice. Lysed cells were centrifuged at 20,000 × *g* for 30 min at 4 °C, and the clarified supernatant was loaded onto 5 mL of Ni Sepharose High-Performance resin (Cytiva) that had been pre-equilibrated with buffer A that lacked ZnCl$_2$. The column was washed with buffer B until protein was no longer observed in the eluent as detected by mixing with Bradford reagent. The target protein was eluted using 5 CVs of buffer B containing 300 mM imidazole. Elutions were pooled and loaded onto 3 mL of Amylose Resin (New England Biolabs) that had been equilibrated with buffer E and incubated for 1 hr at 4 °C. The column was washed with buffer E until no protein was observed in the eluent as detected by Bradford reagent. The target protein was eluted by the addition of 5 CVs of buffer E containing 10 mM maltose. Elutions were pooled, concentrated, and applied to a Superdex 200 Increase 10/300 GL (Cytiva) that had been equilibrated with buffer D. Peak fractions were collected and analyzed using SDS-PAGE. The cleanest fractions were combined and concentrated to 1.0 mg/mL. Concentrated fractions were used immediately, or flash frozen for storage.

### CryoEM sample preparation, data collection, and processing of the mortalin$_{R126W}$-GrpEL1$_{WT}$ complex

Purified mortalin$_{R126W}$ (20 μM), GrpEL1$_{WT}$ (40 μM), and Pam16/Pam18 chimera (12 μM) were incubated at room temperature for 1 hr with occasional mixing. The mixture was applied to a Superdex 200 Increase 10/300 GL (Cytiva) that was equilibrated with buffer D. Fractions consisting of mortalin$_{R126W}$ and GrpEL1$_{WT}$, as visualized by SDS-PAGE, were concentrated to ~0.3 mg/mL. 4 μL of mortalin$_{R126W}$–GrpEL1$_{WT}$ was applied to an UltraAuFoil 1.2/1.3, 300 grid that had been plasma-cleaned using a Gatan Solarus II plasma cleaner (10 s, 15 Watts, 75% Ar/25% O2 atmosphere). Grids were manually blotted for ~6 to 7 s using Whatman No.1 filter paper before vitrification in a 1:1

ethane:propane liquid mixture cooled by liquid $N_2$. The apparatus used was a custom manual plunge freezer designed by the Herzik lab located in a humidified (> 95% relative humidity) cold room maintained at 4 °C.

Data acquisition was carried out at UCSD's CryoEM Facility on a Titan Krios G4 (Thermo Fisher Scientific) operating at 300 keV equipped with a Selectris-X energy filter. Images were collected at a magnification of 130,000x in EFTEM mode (0.935 Å calibrated pixel size) on a Falcon 4 direct electron detector using a 10-eV slit width and a cumulative electron exposure of ~ 60 electrons/Å$^2$. Data were collected automatically using EPU with aberration-free image shifting using a defocus range of − 1.0 to − 2.5 μm. Initially, 952 movies were pruned from 1266 movies using a CTF acceptance range of ~ 3.3–7.1 Å in cryoSPARC Live (cryoSPARC v4.3.1). Patch-based motion correction and CTF estimation were performed in cryoSPARC Live. 255,550 particles were selected using Blob Picker and extracted to a box size of 64 pixels (3.74 Å/pixel). Particles were subjected to two rounds of two-dimensional (2-D) classification (Round 1: 100 classes, 150 Å mask diameter, 200 Å outer mask diameter; Round 2: 50 classes, initial uncertainty factor: 1, 130 Å mask diameter, 160 Å outer mask diameter). Particles resembling a mortalin$_{R126W}$–GrpEL1$_{WT}$ complex were used to generate an ab initio volume (no. of ab-initio classes: 2, maximum resolution: 6 Å, initial resolution: 15 Å, compute per-image optimal scales: on). This model revealed features of an intact mortalin$_{R126W}$–GrpEL1$_{WT}$ complex, which was then used to create templates for template picking. 964,399 particles were identified using template-based particle picking and subjected to three sequential rounds of 2-D classification (50 classes, initial uncertainty factor: 1, 130 Å mask diameter, 160 Å outer mask diameter, no. of final full iterations: 2, batch size per class: 1000). Classes containing false images were removed after iteration 1 while the best-behaving classes were retained in subsequent classifications.

In order to increase particle number and improve views, an additional 1757 movies were collected from the same batch of grids with an effort to include more areas of varying ice thickness. In our datasets, we found that in thinner regions of ice, such as the center of the hole, we observed mostly side-on views of the mortalin$_{R126W}$-GrpEL1 complex, whereas in thicker regions of ice, we observed more top-down views. We rationalize this by the complex's width:height ratio of approximately 1:1.5 with GrpEL1 extended. Therefore, in imaging regions of different thicknesses, we were able to diversify the ratios of side-on and top-down views. These images were subjected to the same patch-based motion correction and CTF estimation in cryoSPARC Live (as described above). 980,189 particles were selected using the previously generated templates and subjected to two sequential rounds of 2-D classification (100 classes, initial uncertainty factor: 1, 120 Å mask diameter, 150 Å outer mask diameter, batch size per class: 1000). 703,658 particles were selected and combined with the previous dataset for a total of ~ 1.35 million particles. These particles were sorted using five sequential rounds of 2-D classification (50 classes, 120 Å mask diameter, 150 Å outer mask diameter, number of final full iterations: 2, batch size per class: 1000). Heterogeneous refinement, using an ab initio volume resembling the mortalin$_{R126W}$–GrpEL1$_{WT}$ complex and a 20S proteasome volume (EMDB-8741), was performed. Particles located in the mortalin$_{R126W}$–GrpEL1$_{WT}$ complex class were re-extracted to a box size of 128 pixels (1.87 Å/pixel) and subjected to two additional rounds of heterogeneous refinement. This first round utilized a mortalin$_{R126W}$–GrpEL1$_{WT}$ volume containing density for the SBD, a mortalin$_{R126W}$–GrpEL1$_{WT}$ volume without density for the SBD, and a 20S proteasome volume (EMDB-8741) (force hard classification: on). The second round utilized the mortalin$_{R126W}$–GrpEL1$_{WT}$ volumes, one containing and one lacking density for the SBD (force hard classification: on). 159,816 particles localizing to the class containing density for the SBD were re-extracted to a box size of 256 pixels (0.935 Å/ pixel). Heterogeneous refinement, using the SBD-containing and SBD-lacking density mortalin$_{R126W}$–GrpEL1$_{WT}$ volumes, was performed, and 132,552 particles classified into the SBD-containing class were selected. A soft mask of the SBD was generated (dilation radius: 7 pixels, soft padding width: 7 pixels) and used in 3-D classification (5 classes, force hard classification: on). Four classes (totaling 121,932 particles) were selected and subjected to heterogeneous refinement using two mortalin$_{R126W}$–GrpEL1$_{WT}$ volumes, one with strong SBD density and one with weak SBD density. 66,065 particles classified into the strong SBD density class were re-extracted to a box size of 384 pixels (0.935 Å/pixel). Reference-based motion correction was performed on these particles using an extensive search for hyperparameters. Non-uniform refinement with CTF refinement turned on yielded a 3.06 Å resolution reconstruction.

To further increase particle number and improve views, an additional 1861 movies were collected as described above and subjected to patch-based motion correction and CTF estimation in cryoSPARC Live (as described above). Particles were identified using template-based particle picking and extracted to a 78-pixel box size (3.74 Å/pixel). Two sequential rounds of 2-D classification (50 classes, initial classification uncertainty factor: 1, 130 Å mask diameter, 160 Å outer mask diameter, batch size per class: 1000) were performed, and 744,973 particles were selected. These particles were subjected to two rounds of heterogeneous refinement (default settings) using one class resembling the full mortalin$_{R126W}$–GrpEL1$_{WT}$ complex and one junk class resembling noise. 248,553 particles classified into the mortalin$_{R126W}$–GrpEL1$_{WT}$ class were re-extracted to a box size of 256 pixels (0.935 Å/pixel). A 3-class ab initio reconstruction was performed on these particles, resulting in 162,368 particles located to a volume resembling the mortalin$_{R126W}$–GrpEL1$_{WT}$ complex. Two rounds of heterogeneous refinement (Round 1: mortalin$_{R126W}$–GrpEL1$_{WT}$ with strong SBD density, mortalin$_{R126W}$–GrpEL1$_{WT}$ with weak SBD density; Round 2: mortalin$_{R126W}$–GrpEL1$_{WT}$ with strong SBD density, noise density) were performed and 103,623 particles classified into the mortalin$_{R126W}$–GrpEL1$_{WT}$ complex volume with strong SBD density were re-extracted to a box size of 384 pixels (0.935 Å/pixel). These particles were subjected to reference-based motion correction using an extensive search for hyperparameters. Particles from both reference-based motion correction jobs (56,380 + 96,395 particles) were combined and subjected to 3-D classification (4 classes). 150,347 particles located to a volume resembling the mortalin$_{R126W}$–GrpEL1$_{WT}$ complex were selected and subjected to 2-D classification (50 classes, initial uncertainty factor: 4, 130 Å mask diameter, 160 Å outer mask diameter, batch size per class: 1000). 138,298 particles were selected and subjected to non-uniform refinement (window dataset (realspace): off, number of extra final passes: 2, initial lowpass resolution: 10 Å, minimize over per-particle scale: on, initialize noise model from images: on, dynamic mask near: 4 Å, dynamic mask far: 10 Å, optimize per-particle defocus: on, optimize per-group CTF params: on, fit spherical aberration: on, fit tetrafoil: on) resulting in a 2.96 Å resolution map.

## CryoEM sample preparation, data collection, and processing of the mortalin$_{R126W}$–GrpEL1$_{Y173A}$ complex

Purified mortalin$_{R126W}$ (20 μM), GrpEL1$_{Y173A}$ (40 μM), and Pam16/Pam18 chimera (12 μM) were incubated at room temperature for 1 hr with occasional mixing. The mixture was applied to a Superdex 200 Increase 10/300 GL (Cytiva) that was equilibrated with a buffer D. Fractions comprising mortalin$_{R126W}$ and GrpEL1$_{Y173A}$, as visualized by SDS-PAGE, were concentrated to ~ 0.5 mg/mL. CryoEM grid preparation was prepared as described for the mortalin$_{R126W}$–GrpEL1$_{WT}$ complex. Data acquisition was carried out at UCSD's CryoEM Facility on a Titan Krios G4 (Thermo Fisher Scientific) operating at 300 keV equipped with a Selectris-X energy filter. Images were collected at a magnification of 165,000x in EFTEM mode (0.735 Å calibrated pixel

size) on a Falcon 4 direct electron detector using a 10-eV slit width and a cumulative electron exposure of ~60 electrons/$Å^2$. Data were collected automatically using EPU with aberration-free image shifting using a defocus range of −1.0 to −2.5 μm. Patch-based motion correction and CTF estimation were performed in cryoSPARC Live and movies were filtered using a CTF acceptance range from 2–6 Å. 266,529 particles were selected using templates generated from the mortalin$_{R126W}$–GrpEL1$_{WT}$ session, extracted to a box size of 96 pixels (3.74 Å/pixel), and subjected to three sequential rounds of 2-D classification (50 classes, initial uncertainty factor: 2, 130 Å mask diameter, 160 Å outer mask diameter, batchsize per class: 200). Three additional rounds of 2D classification were performed using the same parameters with the exception of an initial uncertainty factor of 4. 80,411 curated particles were then subjected to heterogeneous refinement using a mortalin$_{R126W}$–GrpEL1$_{WT}$ ab initio class and a junk class. 41,043 particles localizing to the mortalin$_{R126W}$–GrpEL1$_{WT}$ class underwent two sequential rounds of 2-D classification (25 classes, initial uncertainty factor: 4, 130 Å mask diameter, 160 Å outer mask diameter, number of final full iterations: 2, batch size per class: 10000) and heterogeneous refinement using the mortalin$_{R126W}$–GrpEL1$_{WT}$ ab initio class, a junk class, and a 20S proteasome class (EMDB-8741). 13,325 curated particles were then re-extracted to a box size of 384 pixels (0.735 Å/pixel).

An additional 1995 movies were collected from the same batch of grids and subjected to patch-based motion correction and CTF estimation using cryoSPARC Live. ~1.25 million particles were identified using previously generated templates from mortalin$_{R126W}$–GrpEL1$_{WT}$ and extracted to a box size of 96 pixels (3.74 Å/pixel). Six sequential rounds of 2-D classification (100 classes, initial uncertainty factor: 1, 130 Å mask diameter, 160 Å outer mask diameter, batch size per class: 200) were performed, yielding 552,959 curated particles. These particles underwent heterogeneous refinement (force hard classification: on) using a mortalin$_{R126W}$–GrpEL1$_{Y173A}$ volume generated from the previous heterogeneous refinement, and a junk class. 252,261 particles localized to the mortalin$_{R126W}$–GrpEL1$_{Y173A}$ volume and were subjected to 2-D classification (50 classes, initial uncertainty factor: 4, 130 Å mask diameter, 160 Å outer mask diameter), where 219,184 of the selected particles were subjected to another round of heterogeneous refinement using the same parameters described above. 150,060 particles assigned to the mortalin$_{R126W}$–GrpEL1$_{Y173A}$ class underwent another round of 2-D classification (50 classes, initial uncertainty factor: 4, 130 Å mask diameter, 160 Å outer mask diameter). 130,080 particles were selected and utilized for a 3-class ab initio reconstruction that yielded 69,235 particles resembling mortalin$_{R126W}$–GrpEL1$_{Y173A}$. These particles were re-extracted to a box size of 384 pixels (0.735 Å/pixel). 13,325 and 69,235 unbinned particles from each dataset were combined (82,281) and subjected to non-uniform refinement (window dataset (real-space): off, number of extra final passes: 2, initial lowpass resolution: 10 Å, minimize over per-particle scale: on, initialize noise model from images: on, dynamic mask near: 4 Å, dynamic mask far: 10 Å, optimize per-particle defocus: on, optimize per-group CTF params: on, fit spherical aberration: on, fit tetrafoil: on, fit anisotropic mag.: on), yielding a 3.26 Å resolution EM map.

Our 3.26 Å cryoEM map exhibited anisotropy from dominant views, and thus, we collected an additional 1272 movies at a 10° tilt angle from the same batch of grids. Movies underwent patch-based motion correction and CTF estimation using cryoSPARC Live. Previously generated templates were used to pick 575,214 particles and were extracted to a box size of 96 pixels (3.74 Å/pixel). These particles were subjected to 2-D classification (100 classes, initial uncertainty factor: 1, 130 Å mask diameter, 160 Å outer mask diameter, batch size per class: 200), resulting in 382,150 curated particles. These particles underwent heterogeneous refinement (force hard classification: on) using a mortalin$_{R126W}$–GrpEL1$_{Y173A}$ volume and 20S

proteasome class (EMDB-8741). 212,842 particles assigned to the mortalin$_{R126W}$–GrpEL1$_{Y173A}$ complex class were subjected to an additional round of 2-D classification (50 classes, initial uncertainty factor: 2, 130 Å mask diameter, 160 Å outer mask diameter, batch size per class: 200) where 165,148 selected particles subsequently underwent heterogeneous refinement (force hard classification: on) using a mortalin$_{R126W}$–GrpEL1$_{Y173A}$ class, a 20S proteasome class (EMDB-8741), and a junk class. 87,397 particles localizing to the mortalin$_{R126W}$–GrpEL1$_{Y173A}$ class underwent a 2-class ab intio reconstruction (max resolution: 12 Å, initial resolution 35 Å) where 51,461 particles locating to a volume resembling a mortalin–GrpEL1$_{Y173A}$ complex were selected. These particles underwent heterogeneous refinement (force hard classification: on) using one mortalin$_{R126W}$–GrpEL1$_{WT}$ class with strong SBD density and one mortalin$_{R126W}$–GrpEL1$_{WT}$ class with weak SBD density. 38,765 particles localizing to the complex class with strong SBD density were then re-extracted to a box size of 384 pixels (0.735 Å/pixel).

A soft mask of the SBD was generated (dilation radius: 4.41 pixels, soft padding width: 6 pixels) and used in a focused 3-D classification (2 classes, target resolution: 6 Å, initial structure lowpass resolution: 20 Å) with combined particles from the three mortalin$_{R126W}$–GrpEL1$_{Y173A}$ datasets (13,325 + 69,235 + 38,765). One volume appeared to contain mortalin$_{R126W}$–GrpEL1$_{Y173A}$ with an intact SBD lid (mortalin$_{R126W}$–GrpEL1$_{Y173A}$), whereas the second volume contained mortalin$_{R126W}$–GrpEL1$_{Y173A}$ without the SBD-lid (mortalin$_{R126W}$–GrpEL1$_{Y173A}$-lid). These two volumes, in addition to a junk volume, were used as references for a heterogeneous refinement (force hard classification: on) using the combined particles and yielded 51,324 particles in the mortalin$_{R126W}$–GrpEL1$_{Y173A}$ class and 54,402 particles in the mortalin–GrpEL1$_{Y173A}$-lid class. Non-uniform refinement (window dataset (real-space): off, number of extra final passes: 2, initial lowpass resolution: 10 Å, minimize over per-particle scale: on, initialize noise model from images: on, dynamic mask near: 4 Å, dynamic mask far: 10 Å, optimize per-particle defocus: on, optimize per-group CTF params: on, fit spherical aberration: on, fit tetrafoil: on, fit anisotropic mag.: on) was performed on each of these classes, yielding EM maps with nominal resolutions of 3.38 Å.

### Model building of mortalin$_{R126W}$-GrpEL1$_{WT}$

To model the mortalin$_{R126W}$ NBD and GrpEL1 regions of our mortalin$_{R126W}$–GrpEL1$_{WT}$ complex, we initially referenced the crystal structure of *E. coli* DnaK-GrpE (PDB ID: 1DKG). For the mortalin$_{R126W}$ IDL and SBD, we referenced the crystal structure of the *Hs*Hsp70-SBD bound to a peptide substrate (PDB ID: 4PO2). These models were used as initial structures and docked as rigid bodies into our 2.96 Å EM map. For mortalin$_{R126W}$ we modeled residues 46–639. For GrpEL1-A, residues 59–219 were modeled in addition to a C-terminal serine and alanine representing the linker region. For GrpEL1-B, residues 59–217 were modeled. The bound mortalin$_{R126W}$ substrate was modeled with residues 435–441 representing the interdomain linker. All models were subjected to manual curation and adjustment in COOT, followed by real-space refinement in PHENIX (version 1.21). Model figures were generated in UCSF ChimeraX.

### Model Building of mortalin$_{R126W}$-GrpEL1$_{Y173A}$

**mortalin$_{R126W}$-GrpEL1$_{Y173A}$.** To model our mortalin$_{R126W}$-GrpEL1$_{Y173A}$ structure, we performed rigid-body docking using our mortalin$_{R126W}$-GrpEL1$_{WT}$ model on the mortalin$_{R126W}$ NBD, IDL, GrpEL1-A and GrpEL1-B regions. The SBD and bound substrate regions were docked independently to accommodate the mortalin$_{R126W}$-GrpEL1$_{Y173A}$ SBD EM density. All models were subjected to manual curation and adjustment in COOT, followed by real-space refinement in PHENIX. Model figures were generated in UCSF ChimeraX.

**mortalin$_{R126W}$-GrpEL1$_{Y173A}$-lid.** For the mortalin$_{R126W}$-GrpEL1$_{Y173A}$-lid structure, the lower resolution EM density for mortalin$_{R126W}$'s SBD did not allow for de novo placement of all residues. Therefore, subdomains from the mortalin$_{R126W}$-GrpEL1$_{Y173A}$ structure were rigid body docked, and reference-based refinement parameters in PHENIX were used. All models were subjected to manual curation and adjustment in COOT, followed by real-space refinement in PHENIX. Model figures were generated in UCSF ChimeraX.

## All-atom molecular dynamics (MD) simulations of the mortalin$_{R126W}$-GrpEL1$_{WT}$ and mortalin$_{R126W}$-GrpEL1$_{Y173A}$ complexes using NAMD

**System preparation.** Coordinates for the mortalin$_{R126W}$-GrpEL1$_{WT}$ and mortalin$_{R126W}$−GrpEL1$_{Y173A}$ complexes were defined using the PDB coordinate files following refinement into the corresponding cryoEM maps. These models were solvated with explicit TIP3 water molecules, and the number of Na$^+$ and Cl$^-$ ions was adjusted to neutralize the system charge (with an ionic strength set to 150 mM). The total number of atoms for the final systems was 135,597 for mortalin$_{R126W}$-GrpEL1$_{WT}$ and 138,962 for mortalin$_{R126W}$-GrpEL1$_{Y173A}$ and orthorhombic periodic cells of 88 Å × 129 Å × 127 Å and 89 Å × 125 Å × 132 Å, respectively.

All-atom MD simulations were performed using the CUDA memory-optimized version of NAMD 2.14[56] and CHARMM36m force fields[57]. Each simulation began with an energy minimization step to remove any steric clashes and to ensure that the system was in a low-energy state before dynamic simulation. A minimization of 2000 steps was performed with the following key settings: a cutoff of 12.0 Å for nonbonded interactions, a pair list distance of 14.0 Å, and the use of the Particle Mesh Ewald (PME) method for long-range electrostatics with a grid spacing of 1 Å[58]. Switching was enabled with a switch distance of 10.0 Å, and all atoms were wrapped to the primary simulation cell, including water molecules, to maintain periodic boundary conditions. Following minimization, the system underwent a temperature annealing process. The temperature gradually increased from 60 K to 300 K to allow the system to adapt slowly to the target temperature, reducing the potential for introducing artifacts due to rapid temperature changes. This process was managed through Langevin dynamics with a damping coefficient of 1 ps$^{-1}$, ensuring temperature and pressure stabilization throughout the annealing phase. The system was equilibrated at 300 K. The equilibration phase used the same cutoff, pair list distance, and PME settings as the previous steps to maintain consistency in the treatment of nonbonded interactions. Langevin dynamics continued to control the temperature, with a target pressure of 1.01325 bar maintained via a Langevin piston. The simulation time for this phase was extended to 500,000 steps for both structures to ensure comprehensive equilibration. Harmonic constraints were applied during minimization, annealing, and equilibration to maintain the structural integrity of the starting complexes.

Following minimization, heating, and equilibration, the systems were submitted to productive MD simulations under NPT conditions. A timestep of 2 fs was used, with nonbonded interactions (van der Waals and short-range electrostatic) calculated at each timestep using a cutoff of 12 Å and a switching distance of 10 Å. All simulations employed periodic boundary conditions, utilizing the particle-mesh Ewald method with a grid spacing of 1 Å to evaluate long-range electrostatic interactions. The MD production was carried out for 75,000,000 steps, equivalent to 150 ns, with coordinates saved every 1000 steps (2 ps). Randomized triplicates were run for each structure; 0.45 μs (75,000 frames) combined total for each structure.

**RMSD Calculation.** The root mean square deviation (RMSD) of the GrpEL1-B β-wing, the SBDα lid, the SBDβ subdomain, and the SBDα lid + SBDβ domain from mortalin$_{R126W}$-GrpEL1$_{WT}$ and mortalin$_{R126W}$−GrpEL1$_{Y173A}$ complexes were calculated to assess the structural stability and fluctuations of the individual domains during the simulations. RMSD values were calculated after aligning the trajectory to the initial structure's GrpEL1-B β-wing, SBDα lid, SBDβ subdomain, and SBDα lid + SBDβ domains individually. This was done to remove collective movement to analyze internal flexibility. The results are plotted in Supplementary Fig. 18 for mortalin$_{R126W}$-GrpEL1$_{WT}$, and in Supplementary Fig. 19 for mortalin$_{R126W}$−GrpEL1$_{Y173A}$. The N, O, C, and Cα's of residues 160-217 were used to calculate the GrpEL1-B β-wing, residues 556-639 for the mortalin SBDα lid, residues 440-555 for the mortalin SBDβ subdomain, and residues 440−639 for the SBDα + SBDβ domain. The RMSD analysis was performed using the Visual Molecular Dynamics (VMD) software[59].

RMSD values for the residues involved in defining vectors $v_1$, $v_2$, and $v_3$ (Supplementary Fig. 14) for lateral and medial angular analysis were calculated as described above. RMSD values were computed for the Cα's of Ala569 (mortalin), Glu597 (mortalin), Glu98 (GrpEL1-B), and Lys200 (GrpEL1-B). Results are plotted in Supplementary Fig. 22 for mortalin$_{R126W}$-GrpEL1$_{WT}$, and in Supplementary Fig. 23 for mortalin$_{R126W}$−GrpEL1$_{Y173A}$, to assess the Cα's local motion contribution to the overall vector motion.

**Angular distribution calculation.** To investigate the structural dynamics of mortalin$_{R126W}$-GrpEL1$_{WT}$ and mortalin$_{R126W}$-GrpEL1$_{Y173A}$ during the molecular dynamics simulations, we calculated the angular motion between two predefined vectors for each simulation frame. For medial motions, vector 1 ($v_1$) extended from the Cα atom of residue 569 in the SBDα lid to the Cα atom of residue 597 within the same domain, while vector 2 ($v_2$) spanned from the Cα atom of residue 569 in the SBDα lid to the Cα atom of residue 200 in GrpEL1-B. For lateral motions, vector 1 ($v_1$) was defined from the Cα atom of residue 569 to the Cα atom of residue 597, and vector 3 ($v_3$) connected the Cα atom of residue 569 in the SBD to the Cα atom of residue 98 in GrpEL1-B.

The angle between $v_1$ and $v_2$, and $v_1$ and $v_3$ were determined at each timestep using a custom Tcl script[60] (Angles_Trajectory.tcl) in Visual Molecular Dynamics (VMD) software. The script iteratively updated the coordinates of the selected Cα atoms, calculated the angle between the two vectors, and recorded the angle along with the corresponding frame number. Vectors are visualized in Supplementary Fig. 14.

The angular data was then visualized using MATLAB[61] to depict the trajectory of these angles throughout the simulation. A kernel density estimation was applied to the angular data to generate a smooth distribution curve. To calculate the percentage of the trajectory that explores specific angular ranges in Fig. 7C, D, the angles were monitored throughout the simulations, and the total frames spent within each defined angular range were divided by the total simulation time. The dashed lines in the angle density plots of Fig. 7C, D represent the boundaries between the defined angular regions. These lines demarcate the transitions between different substates, highlighting the specific angular ranges that were used to calculate the percentage of the trajectory spent in each region.

## Elastic network model analysis using ProDy

**System preparation and model construction.** The mortalin$_{R126W}$-GrpEL1$_{WT}$ and the mortalin$_{R126W}$-GrpEL1$_{Y173A}$ structures were parsed using ProDy[62], with secondary structures assigned automatically. Alpha carbon (Cα) selections were made using all Cα in both conformations.

**Anisotropic network model (ANM) calculation.** ANM was used for both conformations to explore the protein's dynamics. The Hessian matrix was constructed to capture the 3D anisotropic behaviors of the protein's movements. A cutoff distance of 10 Å for interactions and a gamma value of 1 indicating the strength of

spring constants were used in the model. The number of modes calculated was set to 10 to focus on the most significant low-frequency motions that correspond to functional movements within the protein. The analysis was conducted with backbone Cα atoms to emphasize the core structural dynamics.

**Mode analysis and visualization.** The ANM modes were analyzed through visualization in VMD using the Normal Mode Wizard (NMWiz) plugin. The ANM model, initially based on Cα atoms, was extended to all atoms to refine the analysis, and include more detailed atomic interactions. This extension was performed using ProDy's[62] extend mode method, which bridges the coarse-grained ANM representation with a detailed all-atom model. This facilitated an intuitive understanding of the modes' physical implications on the protein's structure and function.

All 10 modes were visualized and analyzed for insights into the protein's intrinsic motions. Mean square fluctuations and cross-correlations based on the computed modes were also examined to understand the distribution of movements and the relationship between different regions of the protein.

### Reporting summary

Further information on research design is available in the Nature Portfolio Reporting Summary linked to this article.

## Data availability

The data supporting this study are available from the corresponding author upon request. All models and associated cryoEM maps have been deposited into the Electron Microscopy Data Bank (EMDB) and the PDB. The depositions include final maps, unsharpened maps, half maps, and associated FSC curves. The cryoEM maps have been deposited in the Electron Microscopy Data Bank (EMDB) under accession codes EMD-44675 (Mortalin$_{R126W}$-GrpEL1$_{WT}$); EMD-44676 (Mortalin$_{R126W}$-GrpEL1$_{Y173A}$); and EMD-44677 (Mortalin$_{R126W}$-GrpEL1$_{Y173A}$-lid). The atomic coordinates have been deposited in the Protein Data Bank (PDB) under accession codes PDB: 9BLS (Mortalin$_{R126W}$-GrpEL1$_{WT}$); PDB: 9BLT (Mortalin$_{R126W}$-GrpEL1$_{Y173A}$); and PDB: 9BLU (Mortalin$_{R126W}$-GrpEL1$_{Y173A}$-lid). Source data are provided in this paper. The atomic coordinates of referenced structures are deposited in the PDB under the following codes: 5OBW; 8BG3; 4KBO; 6NHK; 4ANI; 4EZW; 2GUZ; 1DKG; 4PO2; 3N8E; 6ZHI; 4JNF; 4R5L; 4F01. The all-atom molecular dynamics simulations system files and parameter files are available at [https://doi.org/10.5281/zenodo.14014935]. A source data file is available with this manuscript. Source data are provided in this paper.

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

## Acknowledgements

We are grateful to Prof. Rommie Amaro and Prof. Kevin Corbett for providing valuable feedback on this manuscript, and the entirety of the Herzik lab for facilitating insightful discussions. We also thank Dr. Brian Cook for his generous mentorship on protein purification, cryoEM sample preparation, cryoEM data collection and data processing, and preparation of the manuscript. We are grateful to Sam Marchant for producing the R126W point mutation of mortalin. We also thank members of the University of California, San Diego (UCSD)'s Cryo-EM Facility and the broader cryoEM community at UCSD, and Brendan Dennis, Kevin Smith, and the UCSD Physics Computing Facility for their insights and support. Molecular graphics and analyses were performed with UCSF ChimeraX, developed by the Resource for Biocomputing, Visualization, and Informatics at the University of California, San Francisco (UCSF), with support from National Institutes of Health (NIH) grant R01-GM129325 and the Office of Cyber Infrastructure and Computational Biology, National Institute of Allergy and Infectious Diseases. This work was funded by the NIH (R35-GM138206 to M.A.H., T32-GM008326 to M.A.M.) as well as the Searle Scholars Program (M.A.H.).

## Author contributions

M.A.M. and N.I.B. performed molecular biology, sequence alignment, protein preparation, purification, and characterization experiments. M.A.M. performed all cryoEM experiments and analyses under the supervision of M.A.H. Model building and validation were performed by M.A.M. and M.A.H. All experiments and analyses related to anisotropic

network analysis and all-atom molecular dynamics simulations were performed by K.L.M. M.A.M., K.L.M., N.I.B., and M.A.H. contributed to manuscript preparation and editing. Project conceptualization was performed by M.A.M. and M.A.H. Project administration and funding acquisition was performed by M.A.H.

## Competing interests

The authors declare no competing interests.

## Inclusion & Ethics statement

The research described here includes local researchers from the University of California, San Diego. The roles and responsibilities of this research were agreed upon by all included authors.
