## [Transparent Peer Review file · Nature Communications]

Structural insights into GrpEL1-mediated nucleotide and substrate release of human mitochondrial Hsp70

Corresponding Author: Dr Mark Herzik, Jr

Version 0:

Reviewer comments:

Reviewer #1

(Remarks to the Author)

The manuscript is an interesting read and in the main the data look good. However, there are a number of points that should be considered;

Figure 1 could be better designed, panel A requires better description and in its current state is confusing. Scale bars would be useful in C and I am not sure why the data is being introduced in the initial figure which seems to combine background data with the structural data obtained. Panels B,C,D fit with the structural determination theme of the panel but A does not appear to.

Line "mortalinR126w, hereafter referred to as mortalin". This should be clarified as there are several instances of mortalinR126w after this which makes sense as a mutant was used and this should be stated throughout to avoid confusion and mortalinR126W achieves this.

Figure 2 shows the model of the complex but not the data, it would be good to see the map for panel D to show the quality of the density around the nucleotide. It would also show the quality of fit for example in A as well for the interface. The map is rarely shown and example density for all maps should be provided in the Supp material so the reader can assess the quality of the data not just see the models fitted into the data.

Line 300 states that the high resolution enabled modelling of the VLLLDVT peptide but the density is not shown and should be to reflect the quality of the high resolution of the data at this region. The map density in 4A does not look very high resolution.

E.coli should be E. coli, correct throughout.

Section at line 698 To improve angular distribution several more datasets were collected but they appear to be on the same grid, how was this going to change the ratio of views obtained? Was the grid tilted as done for Y173A mutant?

For the validation reports the calculated resolution is consistently lower than the reported resolution, is there a reason for this?

The map for EMD-44677 looks very noisy in the validation reports, why is it contoured to such a low level whereby there is significant noise shown in the map?

Reviewer #2

(Remarks to the Author)

What are the noteworthy results?

The manuscript reports the structure of the Mortalin/GrpE complex. The structure is notable for its use of full-length mortalin in complex with full-length GrpEL1. Prior data in the field was restricted to a crystal structure of the Mortalin nucleotide binding domain modeled with GrpE with the docking based on an existing crystal structure of the Hsp70 nucleotide binding domain in complex with GrpE.

Will the work be of significance to the field and related fields? How does it compare to the established literature? If the work is not original, please provide relevant references.

The work is significant due to the data providing information on full-length mortalin, full-length GrpEL1, and a mortalin substrate. It builds substantially on the previous structural data for this field which was based on docked complexes from isolated crystal structures. Although useful, and although the structure herein is largely similar to the earlier docked models, the confirmation from having full-length proteins used is beneficial for the field. The incorporation of a substrate into the CryoEM samples is highly impactful and a great value-add. It is also notable that the formed complex was made possible by the R126W mutation. This mutation has been studied before, including foundational biophysical assays that provided key insights. The full-length structures here provide even greater information and connect prior biophysical studies to other studies looking at the impact of the R126W mutation on mortalin substrates.

Does the work support the conclusions and claims, or is additional evidence needed?

Yes. The work supports the conclusions and claims. The authors should be commended on the high quality figures that make interpretation of the results easier on the reader. The colors, orientations, and zoom-ins used are excellent for communicating the results.

Are there any flaws in the data analysis, interpretation and conclusions? Do these prohibit publication or require revision?

No. The manuscript appears to be of high quality. The data is analyzed appropriately and the authors provide two possible substrate release mechanisms consistent with the structural and molecular dynamics data.

Is the methodology sound? Does the work meet the expected standards in your field?

The methodology is sound and meets expected standards in the field.

Is there enough detail provided in the methods for the work to be reproduced?

Substantive details are provided throughout the methods section and are detailed enough to allow for readers to reproduce the presented work.

Reviewer #3

(Remarks to the Author)

In their manuscript "Structural insights into GrpEL1-mediated nucleotide and substrate release of human mitochondrial Hsp70", Morizono et al. present a cryo-EM structure of the human mortalin-Grp-EL 1 complex.

I focus here on molecular dynamics simulations (MD) only, as I am neither an expert in cryo-EM and its analysis nor Hsp70.

The authors use MD simulations to investigate the flexibility SBDalpha for the wild type mortalin-GrpEL1 and for the mutant mortalin-GrpEL1_Y173A, where they mutated Y173 in the binding interface between the beta-wing domain of GrpEL1-B and SBDalpha. Cryo-EM structures of the mutant indicate two states. One, where the SBDalpha-lid is bound and ordered and one, where it is unbound and disordered. The authors investigate the flexibility of the SBDalpha lid first using an anisotropic network model (ANM). The analysis of the MD trajectories is motivated by these results.

The simulations themselves seem scientifically sound. The authors did a good job providing the information needed to reproduce their work. I do have some question about technical details. Also, a more comprehensive presentation of the MD results would make it easier for the reader to assess the credibility and implications of the MD results:

1. I have difficulties to reconcile Fig. 7 C and D with the text. The coloring in the figures seems to indicate a concerted motion of SBDalpha from its starting position (0 ns) to its end position (150 ns; selected time point at 130 ns). For mortalin-GrpEL1 in Fig. 7C, this movement goes from about 100 degrees to about 50 degrees for the angle describing the "lateral" motion, as indicated by the colored angle distribution. In the text, however, the authors write "We found that the SBD α rotated up to 50° from its starting position and spent significant time at 20-40° throughout the course of the simulation (Figure 7C, Supplementary Figure 13)." I could not resolve the discrepancy between the text and Figure 7 C. I have a similar problem with the text describing the medial motion of SBDalpha of mortalin-GrpEL1_Y173A and Fig. 7D.

2. I assume that the data in Fig. 7C and 7D (structures and angle distributions) represent selected replicas. I think this should be clarified and the replicas should be specified.

3. The authors have run three replicas for each of the structures. Running replicas is good practice to assess the uncertainty that comes with limited simulation time. In my opinion, the authors could do a better job assessing this uncertainty, which should help the reader to better interpret their results. Also, by better assessing the uncertainty, the reader can better judge if the length of the trajectories is sufficient to justify the conclusions of the authors.

4. For example, it would be useful to present the time series of the lateral AND medial angles for BOTH structures and ALL replicas in the supporting information. Such plots would be especially useful since replica #3 of mortalin-GrpEL1_Y173A

shows both movements (structures at selected time points in Supplementary Figure 15).

5. Similarly, structural stability is much better assessed by aligning individual domains with their initial structure independently and presenting the RMSD for these domains and all replicas as functions of time. The data is already there. I assume that the authors must have looked at these quantities in their analysis. It should be straightforward to add this information to the supplementary information.

6. Along these lines, I do not find the RMSD plots in Supplementary Figure 18 very informative. Quite different structures can have similar RMSD values. This effect is more pronounced for larger RMSD values. Thus, structural difference might be hidden behind large, similar RMSD values. Thus, these plots might be misleading when it comes to structural diversity, which is clearly there.

7. To assess the movement of SBDalpha, the authors have determined angles between vectors, each connecting two Calpha atoms (Supplementary Figure 13). This definition describes domain movement well, but only as long as domains are fairly rigid. I assume that this is the case. If not, however, the angles will also report on internal structural changes of these domains and might be misinterpreted. What is the RMSD of these domains only with respect to the initial structures? How does RMSD for individual domains change with time for the three replicas? After all, the authors' interpretation of cryo-EM data mortalin-GrpEL1_Y173A indicates a state with "disordered" SBDalpha.

8. Have the authors analyzed the interactions of Y173A with SBDalpha in simulations as they have in their cryo-EM structures (Supplementary Figure 9) and compared to AlphaFold2 (Supplementary Figure 17)? Is there anything to learn from such an analysis?

9. Could movies of selected MD trajectories help the reader to better understand the movement, flexibility, and structural stability of SBDalpha?

10. I do not understand line 978 of the manuscript: "The angles were plotted against the step number, with kernel density estimation applied to the data to generate a smooth distribution curve". For kernel density estimation of a distribution, one would not have to plot angles against step number.

Version 1:

Reviewer comments:

Reviewer #1

(Remarks to the Author)

The authors have dealt with the comments and the manuscript looks to ready for publication. Congratulations to them on the work and I look forward to seeing it published.

Reviewer #2

(Remarks to the Author)

The changes made to the manuscript are appropriate relative to my comments in the earlier round of review. I have no additional requested edits.

Reviewer #3

(Remarks to the Author)

Please see attached document.

We thank the reviewers for assessing our manuscript and for providing us constructive feedback and insights. The reviewers' remarks are copied below in gray italics immediately followed by our responses.

Reviewer 1

1. *Figure 1 could be better designed, panel A requires better description and in its current state is confusing. Scale bars would be useful in C and I am not sure why the data is being introduced in the initial figure which seems to combine background data with the structural data obtained. Panels B,C,D fit with the structural determination theme of the panel but A does not appear to.*

We have elaborated on the Figure 1A caption to better describe our representation of Hsp70 catalysis. A scale bar is presented in Figure 1C above the first 2D class represented by a horizontal bar measuring 120Å. Although we recognize the reviewer's comment that background and new data are combined within Figure 1, we believe that panel A of Figure 1 is appropriate in illustrating the current understanding of Hsp70 catalysis, the unknowns associated with co-chaperone interactions, and how our presented work contributes to the overall understanding of Hsp70 biology. We elected to design Figure 1 in such a way that the mortalin and GrpEL1 used in this study can be easily compared across cartoon representations, domain topology, and our cryoEM structures – the former of which are used throughout the manuscript.

2. *Line “mortalin_{R126W}, hereafter referred to as mortalin”. This should be clarified as there are several instances of mortalin_{R126W} after this which makes sense as a mutant was used and this should be stated throughout to avoid confusion and mortalin_{R126W} achieves this.*

We have removed “hereafter referred to as mortalin” and have changed our notation to mortalin_{R126W} throughout the text and figures where appropriate. All changes are highlighted within the updated manuscript.

3. *Figure 2 shows the model of the complex but not the data, it would be good to see the map for panel D to show the quality of the density around the nucleotide. It would also show the quality of fit for example in A as well for the interface. The map is rarely shown and example density for all maps should be provided in the Supp material so the reader can assess the quality of the data not just see the models fitted into the data.*

The cryoEM density for the entire mortalin_{R126W}-GrpEL1_{WT} complex colored by subunit is shown in Figure 1D but we have now included an additional supplementary figure that focuses on the density surrounding the nucleotide binding site that indicates our observed complex is in the apo form. Local resolution analyses for all the maps discussed in this manuscript are also included in the Supplementary Material to indicate the quality of our cryoEM data beyond FSC-estimated resolution.

4. *Line 300 states that the high resolution enabled modelling of the VLLLDVT peptide but the density is not shown and should be to reflect the quality of the high resolution of the data at this region. The map density in 4A does not look very high resolution.*

Our local resolution analysis of the mortalin_{R126W}-GrpEL1_{WT} complex (Supplementary Figure 4) suggests that the resolution at the substrate binding pocket is ~3.2Å. We acknowledge that at this resolution, we cannot unambiguously determine the exact configuration of the bound peptide. Thus, we have reworded line 300 to indicate that our modeled peptide is in agreement with our data (ie. SEC-

MALS, SBD density) and have referenced our local resolution estimation analysis. For transparency, we have included our cryoEM density for the bound substrate into Figure 4C.

5. *E.coli should be E. coli, correct throughout.*

This has been corrected throughout the manuscript.

6. *Section at line 698 To improve angular distribution several more datasets were collected but they appear to be on the same grid, how was this going to change the ratio of views obtained? Was the grid tilted as done for Y173A mutant?*

In our datasets we found that in thinner regions of ice, such as the center of the hole, we observed mostly side-on views of the mortalin_{R126W}-GrpEL1 complex whereas in thicker regions of ice, we observed more top-down views. We rationalize this by the complex's width: height ratio of approximately 1:1.5 with GrpEL1 extended. Therefore, in imaging regions of different thicknesses we were able to diversify the ratios of side-on and top-down views. We have expanded on our rationale in the methods section.

7. *For the validation reports the calculated resolution is consistently lower than the reported resolution, is there a reason for this?*

In the validation reports, the calculated resolution is determined without a mask. This results in a lower calculated resolution and is exacerbated by disordered regions. In the mortalin_{R126W}-GrpEL1_{Y173A} maps, the disorder in the SBD- α subdomains greatly contributes to the worsened calculated resolution. The difference between the unmasked and masked FSC-estimated resolutions is, at least in our experience, consistent with those observed for asymmetric and dynamic complexes.

8. *The map for EMD-44677 looks very noisy in the validation reports, why is it contoured to such a low level whereby there is significant noise shown in the map?*

We acknowledge that the density for the SBD region of this map is poor. The map was contoured such that we could visualize this weak density but do not make strong conclusions about atomistic detail in this region. From this density, we only hypothesize that the SBD- β domain is not substrate bound and may represent a post-substrate release state.

Reviewer 2

1. *What are the noteworthy results?*

The manuscript reports the structure of the Mortalin/GrpE complex. The structure is notable for its use of full-length mortalin in complex with full-length GrpEL1. Prior data in the field was restricted to a crystal structure of the Mortalin nucleotide binding domain modeled with GrpE with the docking based on an existing crystal structure of the Hsp70 nucleotide binding domain in complex with GrpE.

We humbly thank Reviewer 2 for their generous assessment of our manuscript.

2. *Will the work be of significance to the field and related fields? How does it compare to the established literature? If the work is not original, please provide relevant references.*

The work is significant due to the data providing information on full-length mortalin, full-length GrpEL1, and a mortalin substrate. It builds substantially on the previous structural data for this field which was based on docked complexes from isolated crystal structures. Although useful, and although the structure herein is largely similar to the earlier docked models, the confirmation from having full-length proteins used is beneficial for the field. The incorporation of a substrate into the CryoEM samples is highly impactful and a great value-add. It is also notable that the formed complex was made possible by the R126W mutation. This mutation has been studied before, including foundational biophysical assays that provided key insights. The full-length structures here provide even greater information and connect prior biophysical studies to other studies looking at the impact of the R126W mutation on mortalin substrates.

3. *Does the work support the conclusions and claims, or is additional evidence needed?*
Yes. The work supports the conclusions and claims. The authors should be commended on the high quality figures that make interpretation of the results easier on the reader. The colors, orientations, and zoom-ins used are excellent for communicating the results.
4. *Are there any flaws in the data analysis, interpretation and conclusions? Do these prohibit publication or require revision?*
No. The manuscript appears to be of high quality. The data is analyzed appropriately and the authors provide two possible substrate release mechanisms consistent with the structural and molecular dynamics data.
5. *Is the methodology sound? Does the work meet the expected standards in your field?*
The methodology is sound and meets expected standards in the field.
6. *Is there enough detail provided in the methods for the work to be reproduced?*
Substantive details are provided throughout the methods section and are detailed enough to allow for readers to reproduce the presented work.

Reviewer 3

1. *I have difficulties to reconcile Fig. 7 C and D with the text. The coloring in the figures seems to indicate a concerted motion of SBDalpha from its starting position (0 ns) to its end position (150 ns; selected time point at 130 ns). For mortalin-GrpEL1 in Fig. 7C, this movement goes from about 100 degrees to about 50 degrees for the angle describing the "lateral" motion, as indicated by the colored angle distribution. In the text, however, the authors write "We found that the SBD α rotated up to 50° from its starting position and spent significant time at 20-40° throughout the course of the simulation (Figure 7C, Supplementary Figure 13)." I could not resolve the discrepancy between the text and Figure 7 C. I have a similar problem with the text describing the medial motion of SBDalpha of mortalin-GrpEL1_Y173A and Fig. 7D.*

We appreciate the reviewer's commentary and have reworked our representation of the angular analyses in Figure 7C and 7D to improve clarity. The figures focus on the percentage of the trajectory that explores specific angular ranges rather than depicting a linear or concerted motion from one angular region to another over time. The different angles are explored at various times throughout the trajectory, which is why we present the data as a percentage of the total trajectory rather than specifying exact time ranges. The structural snapshots included in these figures are representative of the three primary regions sampled during the simulation, providing a clearer understanding of the most populated substates and angles. We have also included a new Supplemental Figure 14 that details the vectors used for defining and measuring medial and lateral motions.

- I assume that the data in Fig. 7C and 7D (structures and angle distributions) represent selected replicas. I think this should be clarified and the replicas should be specified.*

Figures 7C and 7D specifically use data from replicate 1 of their respective all-atom simulations. Figure 7C represents data from replicate 1 of the mortalin_{R126W}-GrpEL1_{WT} simulation, showing the lateral motion, while Figure 7D shows the medial motion from replicate 1 of the mortalin_{R126W}-GrpEL1_{Y173A} simulation. This is now clearly stated in the figure legend and manuscript.

- The authors have run three replicas for each of the structures. Running replicas is good practice to assess the uncertainty that comes with limited simulation time. In my opinion, the authors could do a better job assessing this uncertainty, which should help the reader to better interpret their results. Also, by better assessing the uncertainty, the reader can better judge if the length of the trajectories is sufficient to justify the conclusions of the authors.*
- For example, it would be useful to present the time series of the lateral AND medial angles for BOTH structures and ALL replicas in the supporting information. Such plots would be especially useful since replica #3 of mortalin-GrpEL1_Y173A shows both movements (structures at selected time points in Supplementary Figure 15).*

We agree with the reviewer that careful analysis of replicates is important for not over-interpreting the data and better assessing uncertainty. To this end, we have included time series plots for lateral and medial angles across all three replicates for both mortalin_{R126W}-GrpEL1_{WT} and mortalin_{R126W}-GrpEL1_{Y173A} in the supplementary information (Supplementary Figures 15 and 16). These plots illustrate the variability and consistency across different simulation runs in an effort to provide a better assessment of uncertainty. Importantly, although the exact angles deviate slightly across replicates, the general trends and our conclusions are consistent.

- Similarly, structural stability is much better assessed by aligning individual domains with their initial structure independently and presenting the RMSD for these domains and all replicas as functions of time. The data is already there. I assume that the authors must have looked at these quantities in their analysis. It should be straightforward to add this information to the supplementary information.*

RMSD analyses for individual domains (GrpEL1-B β -wing, SBD α lid, SBD β domain, and SBD α + SBD β domain) across all replicas are now provided in Supplementary Figures 18 and 19. These figures show the structural stability of each domain throughout the simulations, addressing concerns about potential hidden structural differences.

- Along these lines, I do not find the RMSD plots in Supplementary Figure 18 very informative. Quite different structures can have similar RMSD values. This effect is more pronounced for larger RMSD values. Thus, structural difference might be hidden behind large, similar RMSD values. Thus, these plots might be misleading when it comes to structural diversity, which is clearly there.*

We appreciate the reviewer's feedback regarding the RMSD plots in Supplementary Figure 18. In response to this concern, we have replaced the original RMSD plots with more detailed analyses that

focus on individual domains (GrpEL1-B β -wing, SBD α lid, SBD β domain, and SBD α + SBD β domain) across all replicas. These updated figures, now provided in Supplementary Figures 18 and 19, illustrate the structural stability of each domain throughout the simulations, addressing the concern that structural differences might be hidden behind large, similar RMSD values.

7. *To assess the movement of SBD α , the authors have determined angles between vectors, each connecting two C α atoms (Supplementary Figure 13). This definition describes domain movement well, but only as long as domains are fairly rigid. I assume that this is the case. If not, however, the angles will also report on internal structural changes of these domains and might be misinterpreted. What is the RMSD of these domains only with respect to the initial structures? How does RMSD for individual domains change with time for the three replicas? After all, the authors' interpretation of cryo-EM data mortalin-GrpEL1_Y173A indicates a state with "disordered" SBD α .*

We appreciate the reviewer's insightful comments regarding the potential for internal structural changes within the domains to influence the angle measurements. To address this, we have included detailed RMSD analyses for individual domains (GrpEL1-B β -wing, SBD α lid, SBD β domain, and SBD α + SBD β domain) across all replicas in Supplementary Figures 18 and 19. These figures illustrate the structural stability of each domain throughout the simulations, providing clarity on whether any significant internal structural changes occurred that could affect the angle measurements. Additionally, we have provided RMSD plots specifically for the alpha carbons used in the vector definitions (Supplementary Figures 22 and 23). These plots show the RMSD of these alpha carbons relative to their initial structures to help assess the contribution of the alpha carbon's local motion to the vectors.

8. *Have the authors analyzed the interactions of Y173A with SBD α in simulations as they have in their cryo-Em structures (Supplementary Figure 9) and compared to AlphaFold2 (Supplementary Figure 17)? Is there anything to learn from such an analysis?*

We have included commentary on the interaction between Y173A in GrpEL1-B and the SBD α domain in our MD simulation section. Unfortunately, AlphaFold2 does not predict the GrpEL1-B:SBD α interaction and we therefore cannot draw comparisons.

9. *Could movies of selected MD trajectories help the reader to better understand the movement, flexibility, and structural stability of SBD α ?*

Movies of all MD trajectories are available in the supplementary information (Supplementary Movies 1-8). These movies provide a visual representation of the flexibility and motion observed in the simulations, helping to contextualize the results discussed in the manuscript.

10. *I do not understand line 978 of the manuscript: "The angles were plotted against the step number, with kernel density estimation applied to the data to generate a smooth distribution curve". For kernel density estimation of a distribution, one would not have to plot angles against step number.*

The kernel density estimation was applied to the distribution of angles over the course of the simulation, not directly to the step number. We have clarified this in the manuscript to accurately describe the methodology.

We thank the reviewer for assessing our manuscript and for providing us constructive feedback and insights. The reviewer's remarks are copied below in gray italics immediately followed by our responses.

I thank the authors for addressing the points I had raised. The added data and especially the movies are helpful to better gauge the quality of the simulations, their analysis, and the drawn conclusions.

Please find below one possibly minor issue I noticed in the updated manuscript, which the authors might want to consider. Below that, please find a few additional remarks.

- 1. Watching the movies, I believe that the definition of the vectors to describe lateral and medial motions of SBDalpha picks up some internal flexibility and not only the domain motions, the authors aim to describe. The authors use three residue location to define vectors and the angle between them. The helix, on which one of these residues, GLU597, is located, bends in the case of mortalin_R126W-GrpEL1 and slightly unfolds close to GLU597 at the end in the case of mortalin_R126W-GrpEL1_Y173A. I picked two frames from their movies to illustrate these points (see below). Especially the bending could have a significant effect on their angle distributions. I do not think that this is the movement that the authors had in mind. A simple test/fix would be to move the point to define the vector v1 from GLU597 closer to the center of the helix and before the bend occurs.*

We thank the reviewer for identifying this potential issue. As suggested, we tested our angular analysis using a new vector definition based on residue GLU580 (highlighted in yellow), located just before the helical bend. In the following plots, we compare the lateral and medial motions from our original vector definition (Residue 597, dashed line) with the new vector definition (Residue 580, blue line). The blue and black horizontal lines represent peak standard deviation, the red lines represent standard error, and the yellow dots indicate the starting angular position. Throughout our simulations, the overall trends in lateral and medial motion remain consistent between the two vector definitions. Notably, the sixth column (Degree change), which shows the difference between the average starting position and peak value, demonstrates significant similarity in both direction and magnitude, regardless of the residue used. Any observed differences are likely due to curvature along the alpha-helix axis and greater flexibility near the helix's end, though these factors do not significantly alter the overall motion initially described. Quantification of these analyses are described in the attached table. Across all three replicates, both vector definitions produce, on average, similar angular profiles, effectively capturing the overall motion of the SBD α lid.

mortalin_{R126W}-GrpEL1^{WT}

mortalin_{R126W}-GrpEL1^{Y173A}

Structure	Motion	Residue	Avg Starting (°) ± SD	Avg Peak (°) ± SD	Degree Change (°) ± SD	Observed Motion
mortalin _{R126W} - GrpEL1	Medial Motion	597	94.5 ± 0.33	100.7 ± 8.9	6.2 ± 8.9	away
mortalin _{R126W} - GrpEL1	Medial Motion	580	94.7 ± 0.58	93.9 ± 9.3	0.8 ± 9.3	closer/no sig. change
mortalin _{R126W} - GrpEL1	Lateral Motion	597	90.5 ± 2.1	98.6 ± 4.2	8.1 ± 4.7	away
mortalin _{R126W} - GrpEL1	Lateral Motion	580	84.7 ± 2.1	88.5 ± 9.4	3.8 ± 9.6	away
mortalin _{R126W} - GrpEL1 ^{Y173A}	Medial Motion	597	107.7 ± 0.83	93.0 ± 2.4	14.7 ± 2.5	closer
mortalin _{R126W} - GrpEL1 ^{Y173A}	Medial Motion	580	105.7 ± 1.8	87.1 ± 2.1	18.6 ± 2.8	closer
mortalin _{R126W} - GrpEL1 ^{Y173A}	Lateral Motion	597	95.5 ± 0.97	89.2 ± 4.6	6.3 ± 4.7	closer
mortalin _{R126W} - GrpEL1 ^{Y173A}	Lateral Motion	580	90.3 ± 1.5	85.3 ± 3.1	5.0 ± 3.6	closer

* Independent variables in SD difference assumed

* closer = towards GrpEL1 away = from GrpEL1

Additional remarks

1. *Time stamps in the movies would make their discussion easier.*
2. *In the movies, showing the residues, which define the vectors used for the analysis of lateral and medial motion, as spheres would be helpful.*

We thank the reviewer for their suggestions. In response, we have re-rendered our movies to include both time stamps and spherical representations of the residues defining vectors v1, v2, and v3.

3. *The added RMSD plots do provide additional information for the reader. If I am not mistaken, always the full complex was aligned with the initial structure. To check the stability of individual domains, which in principle can move, separate alignment of these domains makes RMSD values easier to interpret as it removes collective movement of and measures internal flexibility (see SI Figures 18 and 19).*

We thank the reviewer for pointing out a potential miscommunication. In our RMSD analysis (SI Figures 18 and 19), we aligned the trajectory to the GrpEL1-B β -wing, SBD α lid, SBD β subdomain, and the combined SBD α lid + SBD β domains, as previously suggested. We have revised the language in the methods section to clarify that these RMSD plots assess individual domains.

4. *I thank the authors for their explanations of the angle distributions in Fig. 7 and their modifications. In principle, the plots are fine. In my personal opinion, though, the chosen representation sacrifices scientific clarity for aesthetics. The color range in the distributions are not needed. Only three structures are shown. It would be much easier to identify where the corresponding angles are located in the distribution if they were marked by a colored line, for example, and not by one color in a continuous gradient of colors. Also, the curved axis and positioning of the histogram makes it tempting to overinterpret them in context of the structures. The curvature and location of the axis do not add information to the plot. But again, the plots are fine in principle. I personally just find them harder to read than necessary.*

We thank the reviewer for their input. We understand how this representation might be challenging to interpret. However, with the provided annotations, labeling, and additional explanation in the figure caption and methods, we believe the figure remains both informative and aesthetically interesting.

I thank the authors for addressing the points I had raised. The added data and especially the movies are helpful to better gauge the quality of the simulations, their analysis, and the drawn conclusions.

Please find below one possibly minor issue I noticed in the updated manuscript, which the authors might want to consider. Below that, please find a few additional remarks.

1. Watching the movies, I believe that the definition of the vectors to describe lateral and medial motions of SBDalpha picks up some internal flexibility and not only the domain motions, the authors aim to describe. The authors use three residue location to define vectors and the angle between them. The helix, on which one of these residues, GLU597, is located, bends in the case of mortalin_R126W-GrpEL1 and slightly unfolds close to GLU597 at the end in the case of mortalin_R126W-GrpEL1_Y173A. I picked two frames from their movies to illustrate these points (see below). Especially the bending could have a significant effect on their angle distributions. I do not think that this is the movement that the authors had in mind. A simple test/fix would be to move the point to define the vector v1 from GLU597 closer to the center of the helix and before the bend occurs.

Additional remarks

1. Time stamps in the movies would make their discussion easier.

2. In the movies, showing the residues, which define the vectors used for the analysis of lateral and medial motion, as spheres would be helpful.
3. The added RMSD plots do provide additional information for the reader. If I am not mistaken, always the full complex was aligned with the initial structure. To check the stability of individual domains, which in principle can move, separate alignment of these domains makes RMSD values easier to interpret as it removes collective movement of and measures internal flexibility (see SI Figures 18 and 19).
4. I thank the authors for their explanations of the angle distributions in Fig. 7 and their modifications. In principle, the plots are fine. In my personal opinion, though, the chosen representation sacrifices scientific clarity for aesthetics. The color range in the distributions are not needed. Only three structures are shown. It would be much easier to identify where the corresponding angles are located in the distribution if they were marked by a colored line, for example, and not by one color in a continuous gradient of colors. Also, the curved axis and positioning of the histogram makes it tempting to overinterpret them in context of the structures. The curvature and location of the axis do not add information to the plot. But again, the plots are fine in principle. I personally just find them harder to read than necessary.